# High-fidelity spin qubit operation and algorithmic initialization above 1 K

Jonathan Y. Huang[1 ✉], Rocky Y. Su[1], Wee Han Lim[1,2], MengKe Feng[1], Barnaby van Straaten[3], Brandon Severin[1,3], Will Gilbert[1,2], Nard Dumoulin Stuyck[1,2], Tuomo Tanttu[1,2], Santiago Serrano[1], Jesus D. Cifuentes[1], Ingvild Hansen[1], Amanda E. Seedhouse[1], Ensar Vahapoglu[1,2], Ross C. C. Leon[1,8], Nikolay V. Abrosimov[4], Hans-Joachim Pohl[5], Michael L. W. Thewalt[6], Fay E. Hudson[1,2], Christopher C. Escott[1,2], Natalia Ares[3], Stephen D. Bartlett[7], Andrea Morello[1], Andre Saraiva[1,2], Arne Laucht[1,2], Andrew S. Dzurak[1,2 ✉] & Chih Hwan Yang[1,2 ✉]

The encoding of qubits in semiconductor spin carriers has been recognized as a promising approach to a commercial quantum computer that can be lithographically produced and integrated at scale[1–10]. However, the operation of the large number of qubits required for advantageous quantum applications[11–13] will produce a thermal load exceeding the available cooling power of cryostats at millikelvin temperatures. As the scale-up accelerates, it becomes imperative to establish fault-tolerant operation above 1 K, at which the cooling power is orders of magnitude higher[14–18]. Here we tune up and operate spin qubits in silicon above 1 K, with fidelities in the range required for fault-tolerant operations at these temperatures[19–21]. We design an algorithmic initialization protocol to prepare a pure two-qubit state even when the thermal energy is substantially above the qubit energies and incorporate radiofrequency readout to achieve fidelities up to 99.34% for both readout and initialization. We also demonstrate single-qubit Clifford gate fidelities up to 99.85% and a two-qubit gate fidelity of 98.92%. These advances overcome the fundamental limitation that the thermal energy must be well below the qubit energies for the high-fidelity operation to be possible, surmounting a main obstacle in the pathway to scalable and fault-tolerant quantum computation.

To realize the promised benefits of quantum computing, large arrays of qubits will need to operate within densely packed cryogenic platforms, and the heating effects will eventually impose temperatures well above the millikelvin regime[12,14–18]. Spins in semiconductor quantum dots are rising candidates for this undertaking, thanks to their low error rates, long information hold time and industrial manufacturing compatibility[2,3,6,22]. Initial studies of spin qubit operation above 1 K have verified its feasibility, despite suffering from degraded state-preparation-and-measurement (SPAM) and gate fidelities[15–18]. Tackling these challenges requires combining previously unknown device designs and engineering techniques, in areas from initialization to control and readout.

In this work, we operate electron-spin qubits in silicon with SPAM and universal logic fidelities approaching the requirements for surface code error correction[19–21,23]. We enable deterministic two-qubit initialization in silicon above 1 K by an entropy-transferring algorithmic initialization protocol based on two-qubit logic and single-shot readout. The excellent high-temperature performance of semiconductor spin qubits underpins their potential for scalability and integration with classical control electronics. We elaborate on this presenting a complete error analysis in the two-qubit space and characterize every aspect of operation at different temperatures and external magnetic field $B_0$ to open up further studies on error correction and performance of scaled-up systems.

## Device and two-qubit operation

We conduct our study on a prototype two-qubit processor based on a silicon-metal-oxide-semiconductor (SiMOS) double quantum dot (Fig. 1a,b). Each qubit is encoded in the spin state of an unpaired electron[24,25]. The device incorporates multi-level aluminium gate-stacks[26] fabricated on an isotopically enriched $^{28}$Si substrate with 50 ppm residual $^{29}$Si (ref. 27). The quantum dots are electrostatically defined in areas of around 80 nm$^2$ underneath the plunger gates (P1, P2) at the Si/SiO$_2$ interface. An exchange gate (J) controls the inter-dot separation and two-qubit exchange[28–30] at an exponential rate of 20 dec V$^{-1}$. A radiofrequency single-electron transistor (RFSET)[26] operating at 0.21 GHz is used for single-shot charge readout, with a nominal signal integration time $t_{integration}$ = 50 μs. See the Methods for a description of the complete setup. We emphasize that these materials and experimental designs greatly improve the noise performance of our qubits in comparison with previous iterations[16]. For instance, the absence of a micromagnet in the design reduces the coupling of the spin to

[1]School of Electrical Engineering and Telecommunications, University of New South Wales, Sydney, New South Wales, Australia. [2]Diraq, Sydney, New South Wales, Australia. [3]Department of Engineering Science, University of Oxford, Oxford, UK. [4]Leibniz-Institut für Kristallzüchtung, Berlin, Germany. [5]VITCON Projectconsult, Jena, Germany. [6]Department of Physics, Simon Fraser University, Burnaby, British Columbia, Canada. [7]Centre for Engineered Quantum Systems, School of Physics, University of Sydney, Sydney, New South Wales, Australia. [8]Present address: Quantum Motion Technologies, London, UK. ✉e-mail: yue.huang6@unsw.edu.au; a.dzurak@unsw.edu.au; henry.yang@unsw.edu.au

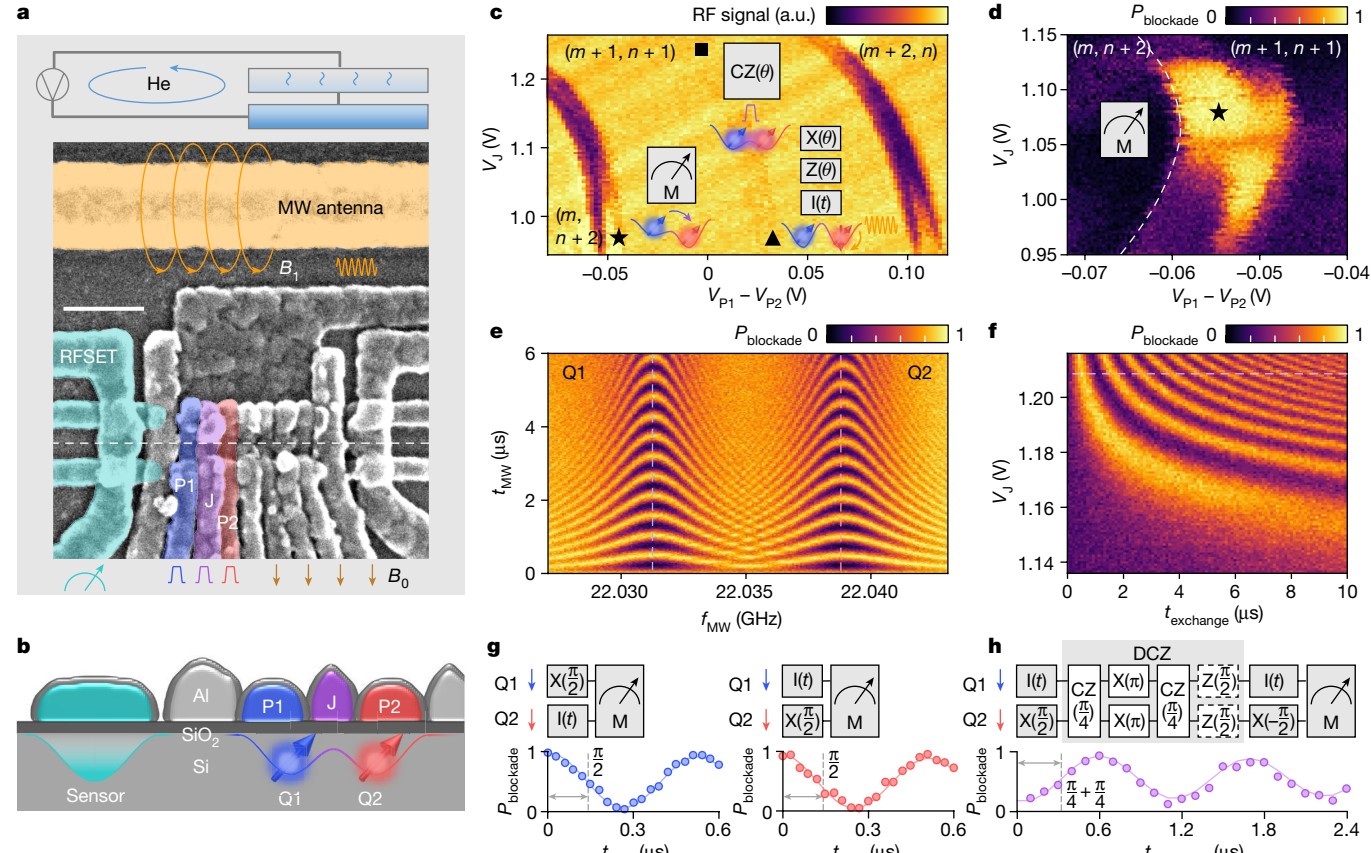

**Fig. 1 | Device and basic operation.** Readout probability is unscaled in all data. **a**, Schematic experimental setup with a scanning electron micrograph of a device nominally identical to that used in this work. The RFSET, the microwave (MW) antenna and other active gate electrodes are highlighted with colours. An external d.c. magnetic field $B_0$ and the antenna-generated a.c. magnetic field $B_1$ are indicated with arrows. The system operates at $T = 1$ K, unless otherwise specified. **b**, Device cross-section schematic with the intended quantum dot location, the electron-spin qubits Q1, Q2 and the RFSET as the charge sensor indicated. **c**, Charge stability diagram as a function of P1, P2 voltage detuning and the J gate voltage $V_J$, showing the operation regime. The operation points for readout (M), single-qubit (X, Z, I) and two-qubit controlled

phase (CZ) operation are labelled as star, triangle and square, respectively. The insets schematically show the operations that are performed at each position. **d**, Probability of reading out a blockaded state, $P_{blockade}$, when preparing $|\downarrow\downarrow\rangle$ and reading out at different $V_J$ and P1, P2 voltage detuning. The readout location for subsequent experiments is set amid the readout window that appears as the high-$P_{blockade}$ region. **e**, Rabi oscillations at $V_J = 1.1$ V as a function of microwave frequency $f_{MW}$ and pulse time $t_{MW}$. **f**, Decoupled controlled phase (DCZ) oscillations as a function of exchange time $t_{exchange}$ and $V_J$. **g**, Calibration of the single-qubit X($\pi$/2) gates. **h**, Calibration of the two-qubit DCZ gate. a.u., arbitrary units. RF, radiofrequency. Scale bar, 100 nm (**a**).

the electric noise generated by thermal fluctuations in dielectrics and metals[31].

To form the qubits (labelled Q1 and Q2), we load an odd number of electrons in the P1 and P2 dots (Fig. 1c). We measure the states using parity readout[32], a type of qubit readout based on Pauli spin blockade (PSB)[33,34]. Charge movement between dots near the inter-dot charge transition point (Fig. 1d) is blockaded when the two qubits are parallel, that is, $|\downarrow\downarrow\rangle$ and $|\uparrow\uparrow\rangle$. We perform readout at the three- to four-electron transition for which the readout window spans 2.5 meV—much larger than the typical valley excitation energies and consistent with the orbital excitation energies previously observed in silicon shell filling (Supplementary Information). After locating the PSB region, algorithmic initialization is used to deterministically prepare a two-qubit state, as introduced later. Single-qubit gates are based on microwave pulses at the electron-spin resonant frequencies ($f_{ESR}$) delivered through the antenna, combined with phase rotations, and two-qubit gates take the form of decoupled controlled phase gate (DCZ)[35,36] (Fig. 1g,h). See the Methods for details on tune-up.

Figure 1e,f shows Rabi and exchange oscillations taken at $T = 1$ K and $B_0 = 0.79$ T. Benefiting from the low charge and spin noise level (Extended Data Fig. 5b–d), feedback on the gate voltage levels, spin

resonance frequencies and microwave amplitudes[9] are not used, which reduces the number of feedback parameters by seven and lowers the time and computation cost. Feedback on the RFSET sensor is retained to automatically maintain the readout signal level over long periods of time[37].

## Initialization and readout

Figure 2a shows the algorithmic initialization protocol to initialize $|\downarrow\downarrow\rangle$ from a mixed state and potentially in the presence of excited states. The resulting two-qubit state composition is verified from the ESR spectrum when the exchange is on. The ESR measurement after the protocol shows only two predominant transitions pertaining to $|\downarrow\downarrow\rangle$, with the amplitude limited by the two-qubit exchange[38]. From these spectra, we extract an initialization fidelity of 99.6%. See the Methods for the protocol details and Supplementary Information for the ESR spectra analysis.

This initialization protocol is robust to low $B_0$, and we expect it to be limited by the fidelities of control and readout on which the protocol relies, and their time scale relative to that of spin relaxation and thermalization. Figure 2a shows the initialization outcomes at $B_0 = 35$ mT, where $k_B T$ is more than 20 times larger than $hf_{qubit}$. The initialization

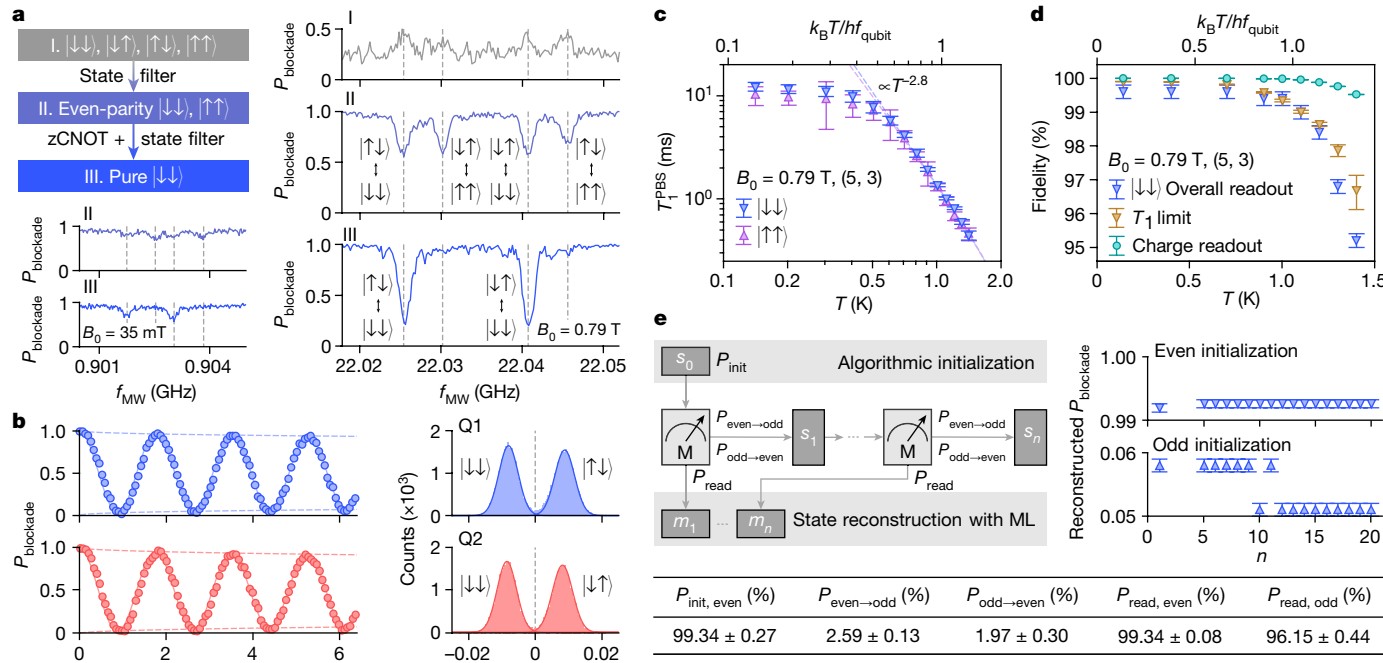

**Fig. 2 | Initialization and readout. a**, Two-qubit algorithmic initialization and the outcomes at $B_0 = 0.79$ T and 35 mT, both at $T = 1$ K in the (5, 3) charge configuration. Stage I represents the conventional ramped initialization over a duration of 100 μs, and with II and III included, we achieve partial and full algorithmic initialization. The traces are taken at $V_J = 1.2$ V, in which the exchange is on, with dashed lines indicating the locations of the four state transitions. **b**, Left, resonantly driven Rabi oscillations of individual qubits for a short $t_{MW}$ and averaged over 500 shots at $B_0 = 0.79$ T and $T = 1$ K in (5, 3). $P_{blockade}$ is unscaled in both traces. Right, corresponding charge readout histograms. The signal integration time $t_{integration}$ is 50 μs. **c**, The lifetime of PSB, $T_1^{PSB}$, for $|\downarrow\downarrow\rangle$ and $|\uparrow\uparrow\rangle$ as a function of temperature from 0.14 K to 1.4 K, at $B_0 = 0.79$ T in (5, 3).

**d**, Measured readout fidelity and estimated $T_1$-limited spin readout fidelity of $|\downarrow\downarrow\rangle$, and charge readout fidelity as a function of temperature from 0.14 K to 1.4 K, at $B_0 = 0.79$ T in (5, 3) with $t_{integration} = 50$ μs. **e**, State reconstruction and state-preparation-and-measurement (SPAM) error analysis using repeated PSB readout at $B_0 = 0.79$ T and $T = 1$ K in (5, 3). We initialize $s_0 = |\downarrow\downarrow\rangle$ using the algorithmic initialization or $|\uparrow\downarrow\rangle$ by π-pulsing on Q1 after the algorithmic $|\downarrow\downarrow\rangle$ initialization. We then perform $n$ PSB readouts, through which the state evolves into $s_n$. Finally, we apply machine learning (ML) on the readout outcomes $m_1$–$m_n$ to extract the initialization, readout and spin-flip probabilities and reconstruct the states. The results are shown in the table and the plots. Error bars represent the 95% confidence level.

fidelity remains above 99% at $B_0 = 85$ mT and above 90% at $B_0 = 35$ mT, but the ESR amplitude is further reduced because of the deviation from parity readout with the small Zeeman energy difference d$E_Z$ (see also Extended Data Fig. 6f). In most operating conditions, the protocol takes around three iterations, and the initialization process spans between 100 μs and 200 μs (Extended Data Fig. 3c,d).

When addressing individual qubits without pulsing on the J gate, we obtain Rabi oscillations with a raw amplitude of 0.950–0.966 for the two qubits (Fig. 2b) at $T = 1$ K and $B_0 = 0.79$ T. We see that both Rabi oscillations start from 0.996 and go down to 0.030–0.046 after a π duration. We thus estimate that the initialization and the overall readout fidelities are 99.6% for $|\downarrow\downarrow\rangle$, and at least 95.0% for $|\downarrow\uparrow\rangle$ and $|\uparrow\downarrow\rangle$.

At $T = 1$ K and $B_0 = 0.79$ T, the relaxation time $T_1$, which is the time for a single spin flip at the single-qubit operation point, is 12.23 ± 2.11 ms and the PSB relaxation time $T_1^{PSB}$, which is the lifetime of a blockaded state at the PSB region, is 1.36 ± 0.06 ms. We use $t_{integration} = 50$ μs, a time much shorter than $T_1^{PSB}$, to achieve a charge readout fidelity of 99.7%. With these considered, the Rabi amplitude is most probably limited by control errors and the diabaticity in reading out odd-parity states.

Figure 2c,d shows the temperature dependence of these metrics between 0.14 K and 1.4 K at $B_0 = 0.79$ T. The PSB relaxation times scale with $T^{-2.8}$ above 0.5 K, dropping by tenfold to 0.45 ms at $T = 1.4$ K. This reduction implies that future readout techniques should avoid compromising on the total readout time. Below 1 K, the overall readout fidelity for $|\downarrow\downarrow\rangle$ falls slowly and seems to be limited by neither $T_1^{PSB}$ nor charge readout, whereas above 1 K, these two limitations are present. $T_1$-induced errors increase at a higher rate than charge readout errors

and seem to be the dominating factor towards even higher temperatures. Overall, SPAM around 1 K is comparable to that at millikelvin temperatures and remains workable at least until 1.4 K.

Finally, we test repeated parity readout at $T = 1$ K. We apply machine learning to infer the parity readout errors and probabilities during SPAM and reconstruct the true initial state parity using the cumulative readout outcomes[39]. Figure 2e shows this protocol along with the results of our analysis. See the Methods for details on the machine learning approach. The SPAM fidelities are captured by $P_{init}$ and $P_{read}$, and the probabilities of state changes during each readout cycle are captured by $P_{even\rightarrow odd}$ and $P_{odd\rightarrow even}$. With the algorithmic $|\downarrow\downarrow\rangle$ initialization and using 20 readout cycles ($n = 20$), we infer $P_{init,even}$, $P_{init,odd}$ to be 99.34 ± 0.27%, 94.67 ± 0.73%, and $P_{read,even}$, $P_{read,odd}$ to be 99.34 ± 0.08%, 96.15 ± 0.44% respectively. For the $|\downarrow\downarrow\rangle$ initialization with $n = 5$, the reconstructed $P_{blockade}$ increases from 99.2% to 99.3%, and for the $|\uparrow\downarrow\rangle$ initialization with $n = 12$, $P_{blockade}$ decreases from 5.8% to 5.1%. The full set of probabilities are detailed in the Supplementary Information.

## Single-qubit performance

Relaxation time ($T_1$) and dephasing time ($T_2$) as well as the single-qubit control fidelities tend to be notable in silicon[40–42], with single-qubit fidelities around 99% previously attained at $T > 1$ K (refs. 16,17). The improvements in device materials and design further improved the performance of individual qubits.

We first study $T_1$ and $T_2$ (Fig. 3a,b) in this device in the (1, 3) and (5, 3) charge states near the optimal $B_0$. These regimes are expected to have similar relaxation mechanisms as (3, 3) (ref. 43), which is studied in

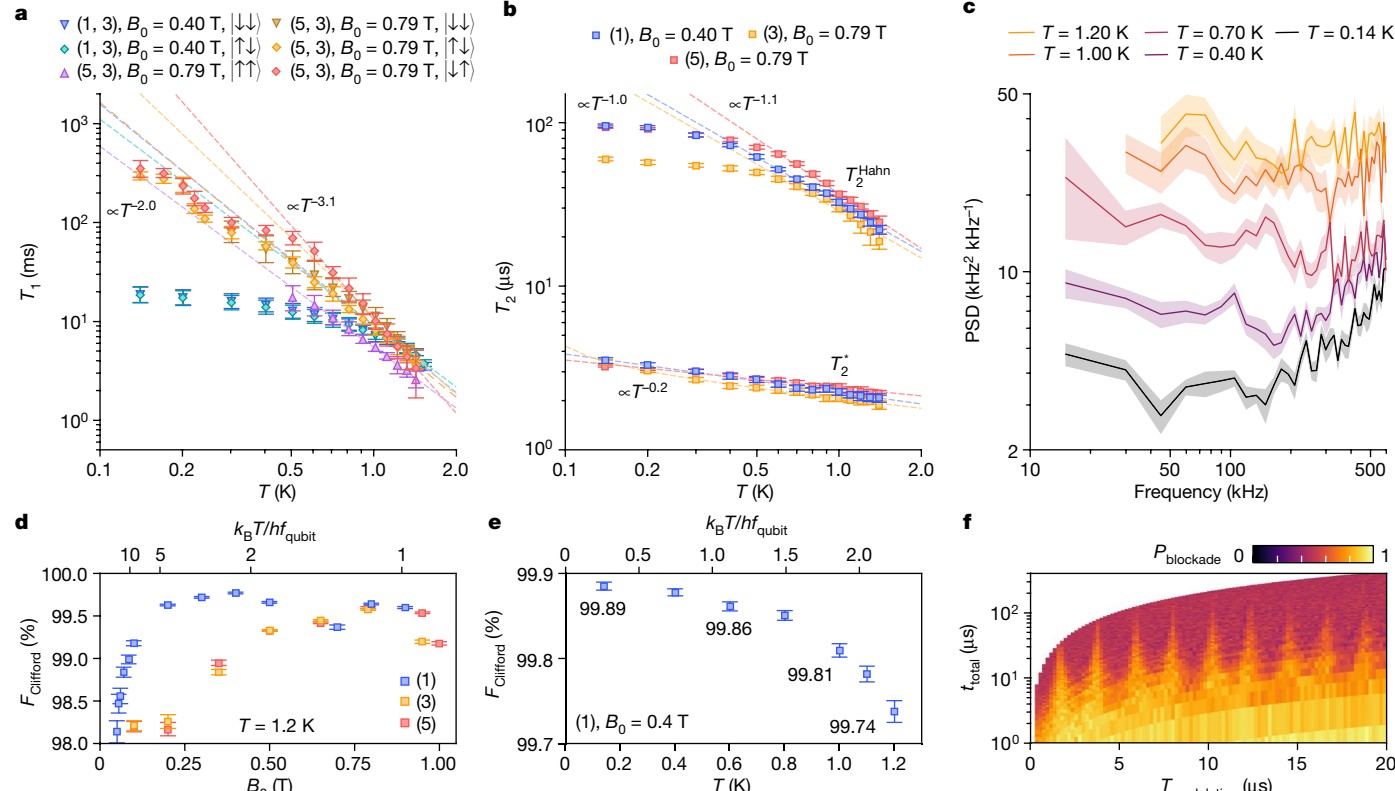

**Fig. 3 | Single-qubit performance. a**, Relaxation time, $T_1$, of different states as a function of temperature from 0.14 K to 1.4 K in different charge configurations for Q1, Q2. **b**, Dephasing times, $T_2^*$ and $T_2^{Hahn}$, as a function of temperature from 0.14 K to 1.4 K in different charge configurations for Q1. **c**, Single-qubit noise PSD of Q1 based on the CPMG protocol[44,45] at different temperatures at $B_0 = 0.79$ T. **d**, Single-qubit Clifford gate fidelity as a function of $B_0$ from 50 mT to 1 T at $T = 1.2$ K in different charge configurations for Q1. **e**, Single-qubit Clifford gate fidelity as a function of temperature at $B_0 = 0.4$ T with the one-electron configuration. **f**, Demonstration of a dressing protocol, the SMART protocol on Q1 (ref. 8) at $B_0 = 0.5$ T and $T = 1$ K with $f_{Rabi} = 1.44$ MHz, showing periodical modulation optima. Error bars represent the 95% confidence level.

ref. 16. However, the absence of a micromagnet in the present study affects some of the physical mechanisms of relaxation and decoherence. The dominating relaxation contributors—charge noise, Johnson noise, Orbach and Raman phonon scattering, and their coupling to the qubits—may change at different temperature ranges, giving rise to an intricate temperature dependence of $T_1$ (ref. 15). Moreover, the parity readout convolutes the relaxation and thermal excitation of both qubits. Nevertheless, we note that the temperature dependence of $T_1$ shown in Fig. 3a falls between $T^{-2.0}$ and $T^{-3.1}$ above 1 K. Moreover, the thermal equilibrium shifts from $|\downarrow\downarrow\rangle$ when $k_B T \ll hf_{qubit}$ to a mixed state when $k_B T \geq hf_{qubit}$. See Extended Data Fig. 4 for a more detailed $T_1$ analysis. The temperature dependence of $T_2^{Hahn}$ in different configurations falls between $T^{-1}$ and $T^{-1.1}$, whereas $T_2^*$ scales uniformly to $T^{-0.2}$. The temperature scaling power of both $T_1$ and $T_2$ are lower than those in most of the previous results[15,16,18]. We expect that the more purified silicon and the absence of a micromagnet in this study affect some of the physical mechanisms of relaxation and decoherence. The bias of Z errors (dephasing noise) to X errors (depolarization noise) can be indicated by the $T_1/T_2$ ratio shown in Extended Data Fig. 5a.

We perform noise spectroscopy based on a Carr–Purcell–Meiboom–Gill (CPMG) protocol[44,45], which uses a single qubit as a noise probe to determine the noise power spectral density (PSD) at different temperatures. As shown in Fig. 3c, the overall noise level rises with temperature within the detectable frequency range. In the white noise regime (<200 kHz), the noise power spectral density increases by an order of magnitude from 0.14 K to 1.2 K. We notice an increase in the apparent PSD at higher frequencies that we characterize as blue noise, possibly because of accumulated microwave pulse miscalibration, or an effect

from the high-power driving[9,46]. See Extended Data Fig. 5e for the full set of power noise spectral density traces, and Extended Data Fig. 5f for another measurement of the microwave effect.

The optimized single-qubit Clifford fidelity in randomized benchmarking[47] is up to 99.85 ± 0.01% (see Supplementary Information for data). Correction of crosstalk is crucial because of the relatively small difference in $f_{ESR}$ in this device. See the Methods for crosstalk correction and the implementation of randomized benchmarking. As shown in Fig. 3d, we observe a fidelity reduction at low $B_0$, limited by crosstalk (Extended Data Fig. 7c), or near an excited state degeneracy at high $B_0$ in which dephasing is enhanced by spin–orbit coupling[48]. Even when $k_B T \approx 7hf_{qubit}$, we still measure larger than 99% fidelities. The qubits are operable with distinguishable frequencies at $B_0$ as low as 25 mT, at which point all operation protocols must be revisited[4]. See Extended Data Fig. 6 for the full study. These results suggest the possibility of ultralow $B_0$ operation to markedly reduce the hardware and power cost[49].

Furthermore, we extend the recent demonstration of a dressing protocol, the SMART protocol, for a single qubit from 0.1 K (refs. 4,5,8) to 1 K. See Extended Data Fig. 7e for the gate sequence. This demonstration substantiates the potential to continuously drive a large number of spin qubits with a global field above 1 K in future architectures.

## Two-qubit performance

Two-qubit gate fidelities in silicon have recently reached the fault-tolerant requirements[23,36,50–52], and extending this to above 1 K becomes of great interest. We perform a decoupled controlled phase (DCZ) operation[52], which incorporates a decoupling X(π) gate on individual qubits in the middle of the CZ gate to extend coherence and cancel

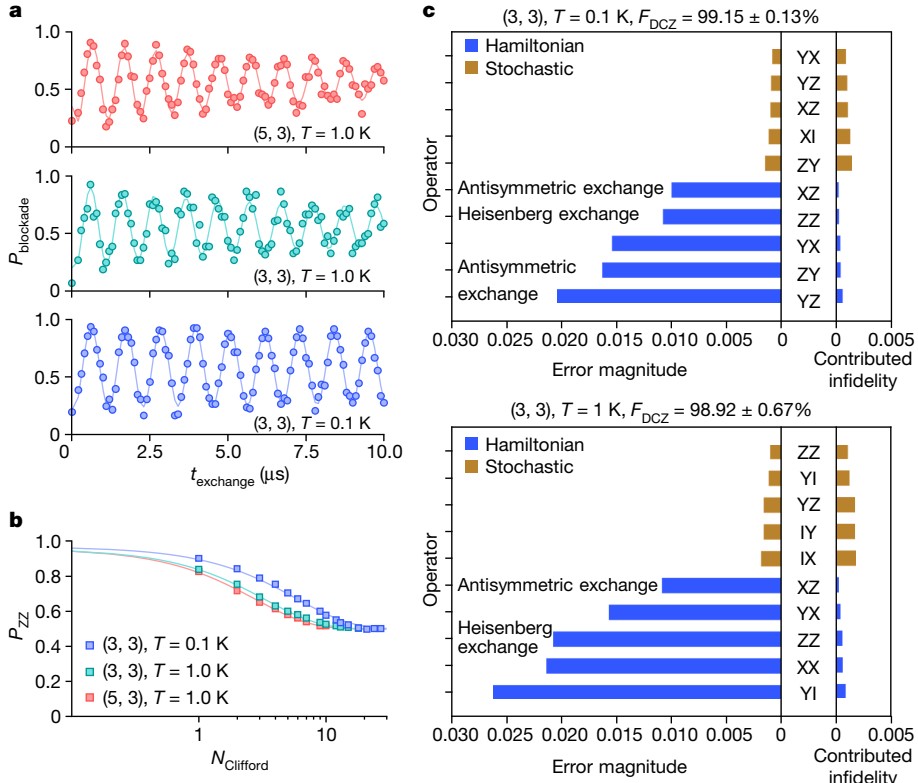

**Fig. 4 | Two-qubit performance. a**, DCZ oscillations at $B_0 = 0.79$ T, $T = 0.1$ K and 1 K. The visibility is limited by microwave-induced noise[46], J gate pulsing, and the use of only partial algorithmic initialization. **b**, Two-qubit randomized benchmarking at $B_0 = 0.79$ T, $T = 0.1$ K and 1 K. **c**, Breakdown of error channels using pyGSTi (ref. 59), based on the final FBT error generators at $B_0 = 0.79$ T, $T = 0.1$ K and 1 K, with the error magnitudes plotted towards the left and their contributed infidelities plotted towards the right. Each sub-figure includes the five largest contributing channels for both Hamiltonian (blue) and stochastic (gold) errors, respectively. We note that Hamiltonian errors contribute to the infidelity in second order, but stochastic errors contribute in first order[60]. Common error channels are labelled with their physical interpretations. Error bars represent the 95% confidence level.

Stark shift-induced phase errors[35,36]. The quality factor of the DCZ oscillation is well above 100 at $T = 0.1$ K and remains at least 50 above 1 K, exhibiting a coherence reduction similar to that in $T_2^{\text{Hahn}}$ (Fig. 4a). See Extended Data Fig. 8 for the full characterization of exchange.

We assess the DCZ gate metrics using two-qubit interleaved randomized benchmarking (IRB)[50] and fast Bayesian tomography (FBT)[53] (Methods), which report DCZ fidelities of 99.8 ± 0.2%, 99.15 ± 0.13% at $T = 0.1$ K, and 97.7 ± 1.5%, 98.92 ± 0.67% at $T = 1$ K. In Fig. 4c, we show the five largest components from Hamiltonian error and stochastic error obtained from FBT. Going from 0.1 K to 1 K, we see a change in the error landscape. At both temperatures, exchange noise appears as one of the main noise sources. We observe terms represented by the Heisenberg exchange and the antisymmetric exchange, also known as Dzyaloshinskii–Moriya interaction. We expect that the antisymmetric exchange leads to Hamiltonian error terms that couple in a ZY- or ZX-like manner. Although this is an important source of error that should be reduced, we note that the main contributions to the infidelity itself come in the form of stochastic errors that contribute linearly to the infidelity. See the Supplementary Information for details.

The detailed nature of the dominant error processes in silicon spin qubits offers a lot of opportunities for innovations in codes and architectures. We observe a bias in the error rates towards dephasing, generally larger than 100:1 up to at least $T = 1.5$ K, for which increased fault-tolerant thresholds are possible[54–56]. To exploit such gains, further research would be needed to characterize the process of error syndrome extraction, in which each cycle involves SPAM on the ancilla qubits during which the data qubits can undergo decoupling. We expect a moderate decrease in the noise bias from decoupling with increasing temperatures (Extended Data Fig. 5a), but this may not be true for even higher temperatures. The CZ-type operation we use as a 2-qubit gate can be bias-preserving, but fully exploiting this bias for QEC will require syndrome extraction circuit design to avoid injection of spin-flip errors from SPAM of the ancilla qubits into the data qubits.

## Outlook

The use of algorithmic qubit initialization and the realization of high-fidelity universal logic in this work bring SiMOS spin qubits at temperatures above 1 K into the realm of fault tolerance. Furthermore, the proven ability to operate at low $B_0$ will benefit large-scale global control[7,10] with low driving frequency and reduce the cost of microwave instrumentation. This further strengthens semiconductor spin qubits as an affordable approach. Apart from setting the benchmark for initialization, control and readout fidelities at elevated temperatures, we present here a complete study of the properties of the two-qubit system (metrics summarized in Extended Data Table 1). We show certain robustness against the charge configuration and the applied magnetic field above $T = 1$ K, which is important for large-scale operation. The similar temperature dependence of $T_1$ and $T_2$ in different configurations above $T = 1$ K suggests a potentially weaker effect from qubit variability[30] at such temperatures.

Challenges remain in raising SPAM and control fidelities to far above 99% to achieve truly fault-tolerant operation. We find that the control process potentially injects errors into the spin readout, which should be addressed to increase the readout fidelity. In the future, incoherent errors can be ameliorated by improving the quality of the Si/SiO₂ interface and the SiO₂ layer and reducing the noise level in the experimental setup. We expect that the fabrication of SiMOS devices in industrial foundries[6,22] will bring a reduction in defects and charge impurities[57,58],

which will increase qubit coherence times and decrease the required feedback. A faster readout is also desired to reduce the initialization time and consequently the overall SPAM duration.

Ultimately, the scalability of spin qubits will rely on scalable control techniques, such as the multi-qubit SMART protocol[4,5,8], in which the qubits are continuously driven by a modulated microwave field. In such schemes, the driving pulses decouple the qubits from noise and eliminate free precession, during which they are most sensitive to decoherence in the system. Advanced shaping of control pulses can also account for coherent errors arising from miscalibration and parameter drifts.

The engineering challenges in building a fault-tolerant, million-qubit quantum processor remain formidable. One of the most promising pathways to solve them will be the adoption of successful CMOS chip manufacturing methods. The results presented here show that high-fidelity quantum operations can be achieved in a CMOS-compatible silicon processor, at high enough temperatures to realistically permit the operation and integration of classical control circuits, making a scalable semiconductor quantum processor a plausible reality in the future.

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

# Article

## Methods

### Measurement setup

The full experimental setup is shown in Extended Data Fig. 1. The device is measured in a Bluefors XLD400 dilution refrigerator. The device is mounted on the cold finger. Within $T = 1$ K, elevation from the base temperature is achieved by switching on and tuning the heater near the sample. Temperatures above 1 K are attained by reducing the amount of He mixture in the circulation and consequently the cooling power. Temperature control becomes non-trivial above 1.2 K and nonviable above 1.5 K.

An external d.c. magnetic field is supplied by an American Magnetics AMI430 magnet. The magnetic field points in the [110] direction of the Si lattice. The d.c. voltages are supplied with Basel Precision Instruments SP927 LNHR DACs through d.c. lines with a bandwidth of 0–20 Hz. Dynamic voltage pulses are generated with a Quantum Machines OPX and combined with d.c. voltages by custom voltage combiners at the 50 K stage in the refrigerator. The OPX has a sampling time of 4 ns. The dynamic pulse lines in the fridge have a bandwidth of 0–50 MHz, which translates into a minimum rise time of 20 ns. Microwave pulses are synthesized using a Keysight PSG8267D Vector Signal Generator with the baseband I/Q and pulse modulation signals from the OPX. The modulated signal spans from 250 kHz to 44 GHz but is band-limited by the fridge line and the d.c. block.

The charge sensor comprises a single-island SET connected to a tank circuit for reflectometry measurement. The return signal is amplified by a Cosmic Microwave Technology CITFL1 LNA at the 4 K stage and a Mini-circuits ZX60-P33ULN+ LNA followed by two Mini-circuits ZFL-1000LN+ LNAs at room temperature. The Quantum Machines OPX generates the tones for the RFSET and digitizes and demodulates the signals after the amplification.

### Device tune-up

We first load the electrons according to the mapping of the double-dot charge configurations over a large range, using lock-in charge sensing measurement[61] with the RFSET. The measurement can be done in the physical gate basis by sweeping $V_{P1}$ and $V_{P2}$, as shown in Extended Data Fig. 2a, or in the virtual gate basis by sweeping $V_{P1} - V_{P2}$ and $V_J$, as shown in Fig. 1a. In the virtual gate basis, voltages of $-0.32\,V_J$ and $-0.25\,V_J$ are applied on P1 and P2 to compensate for the effect of pulsing J. During operation, each dot is loaded with an odd number of electrons, from which the unpaired electron carries the spin information. This is denoted as the $(m + 1, n + 1)$ charge state in the charge maps, where $m$ and $n$ are even numbers.

The tune-up proceeds with locating the PSB region around the inter-dot charge transition, as indicated by the dashed square in Extended Data Fig. 2c,d. The initial PSB search involves loading a mixed spin state in $(m + 1, n + 1)$, which has some probability of being even-parity ($|\downarrow\downarrow\rangle$ or $|\uparrow\uparrow\rangle$), and subsequently pulsing to a location near the inter-dot charge transition point. Single-shot charge readout is performed before and after reaching the location and the final readout signal is provided by subtracting the two signals. Except at ultra-low $B_0$, the readout mechanism is dominated by parity readout because of the relatively large $dE_Z$ between the two qubits[32]. An even-parity spin state appears as blockaded in the PSB region, which translates to a lower radiofrequency signal compared with that from an unblockaded state. The averaged radiofrequency signal, therefore, indicates the probability of having an even-parity state across multiple shots.

The two-level behaviour in the PSB region is used to perform single-shot spin readout. The readout signal in each shot of the experiment is compared with a preset threshold that lies between the two levels, as we see in the readout histograms in Fig. 2b. We assign value 1 to a blockaded readout, and value 0 to an unblockaded readout. Finally, we average over all shots to obtain $P_{blockade}$ for the statistics.

Extended Data Fig. 2f shows the ESR spectrum as a function of $V_J$, in which we identify two regimes. At $V_J < 1.175$ V, only two transitions pertaining to the driven rotation of the individual qubits are detected. Driven over time, these transitions correspond to the Rabi oscillations in Fig. 1e. At $V_J > 1.175$ V, in which the exchange energy is large, we see four transitions among the four two-qubit states corresponding to the controlled rotation operations[38,50]. The layout of the transitions, together with the background signal, shows the composition of the initialized qubit state. The traces in Fig. 2a are taken from these measurements at high $V_J$. A more scalable two-qubit operation is the electrically pulsed controlled phase operation (CZ)[35,36]. This is adopted in this work to construct the CZ gate (Extended Data Fig. 2g), or the DCZ gate in the main text.

### Algorithmic initialization

When the qubit energy $hf_{qubit}$ is greater than the thermal energy $k_BT$, electron-spin qubit initialization may rely on intrinsic polarization mechanisms such as spin-dependent tunnelling from a reservoir[62–64], PSB[17,18,49,65] or relaxation[16,43,66]. Higher-fidelity single-qubit state preparation can be achieved using initialization by measurement[39,67] and conditional single-qubit pulses[9,68]. These approaches either partially rely on intrinsic polarization or require readout with a reservoir, which are incompatible with operation at elevated temperatures. In this work, we design a generic two-qubit algorithmic initialization protocol that works in conditions for which $hf_{qubit}$ is comparable to or less than $k_BT$. The method is applicable to a large-scale qubit array, in which initialization and readout are performed pairwise[16,49,65].

The algorithmic initialization protocol, as shown in Extended Data Fig. 3a, proceeds as follows:

1. Enter $(m + 1, n + 1)$ to create two unpaired spins in the double-dot system.
2. This results in one of the $|\downarrow\downarrow\rangle$, $|\downarrow\uparrow\rangle$, $|\uparrow\downarrow\rangle$ and $|\uparrow\uparrow\rangle$ states. The probability of creating the ground state $|\downarrow\downarrow\rangle$ decreases as the temperature increases, as the thermal energy becomes comparable or greater than the qubit exchange coupling and the Zeeman energies.
3. Ramp to the PSB region for parity readout and apply a filter that rejects odd-parity states. The parity readout preserves the even-parity states as long as it is performed faster than the spin relaxation time[32].
   (a) If the state is unblockaded and thus determined as an odd-parity ($|\downarrow\uparrow\rangle$, $|\uparrow\downarrow\rangle$) or excited state, the initialization is restarted.
   (b) If the state is blockaded and thus determined as even-parity ($|\downarrow\downarrow\rangle$, $|\uparrow\uparrow\rangle$), the initialization proceeds to the next stage.
4. This results in either $|\downarrow\downarrow\rangle$ or $|\uparrow\uparrow\rangle$, with an increased probability of $|\downarrow\downarrow\rangle$ from step 3. We calibrate the CZ gate at this stage, either from the exchange-induced splitting of the ESR transitions (Extended Data Fig. 2f) or from the CZ oscillations (Extended Data Fig. 2g).
5. A zero-CNOT (zCNOT) gate[23] is performed to convert $|\uparrow\uparrow\rangle$ into $|\uparrow\downarrow\rangle$, leaving $|\downarrow\downarrow\rangle$ unchanged. The construction of the zCNOT gate in this work is shown in Extended Data Fig. 2g.
6. Ramp to the PSB region for parity readout, and apply a filter that rejects odd-parity states.
   (a) If the state is unblockaded and thus determined as $|\uparrow\downarrow\rangle$ or an excited state, the initialization is restarted.
   (b) If the state is blockaded and thus determined as $|\downarrow\downarrow\rangle$, the initialization is determined to be completed.
7. The resulting state is purely $|\downarrow\downarrow\rangle$.

The if conditions above are implemented using real-time logic in the FPGA.

The protocol can also be adapted to prepare any other state on the parity basis. $|\uparrow\downarrow\rangle$ and $|\downarrow\uparrow\rangle$ can be prepared from $|\downarrow\downarrow\rangle$ with a microwave π pulse on Q1 and Q2. $|\uparrow\uparrow\rangle$ can be prepared by replacing the zCNOT with CNOT in the algorithm.

We test the algorithmic initialization in a wide range of $B_0$ from 1 T down to 25 mT. The results at different stages of the protocol are seen in Fig. 2a and Extended Data Fig. 3b. Stage I, the outcome of a 100-µs ramp into the operation point, has a mixture of $|\downarrow\downarrow\rangle$, $|\downarrow\uparrow\rangle$, $|\uparrow\downarrow\rangle$ and $|\uparrow\uparrow\rangle$ states and the measured ESR transitions are almost indistinguishable. After Stage II, the output is a mixture of $|\downarrow\downarrow\rangle$ and $|\uparrow\uparrow\rangle$ with the odd-parity states or excited states filtered out through PSB, which can be identified from the associated ESR transitions. Stage III converts the remnant $|\uparrow\uparrow\rangle$ into $|\uparrow\downarrow\rangle$, which is then filtered out through PSB because of the odd parity.

It is also important to assess the time cost for the algorithmic initialization, as it involves multiple control and readout iterations. The table in Extended Data Fig. 3b breaks down the time spent on control and readout. We see that the readout integration time $t_{\text{integration}}$ dominates the time consumption. At $B_0 = 85$ mT and $T = 1$ K, the full algorithmic initialization takes an average of around three iterations, which totals around 150 µs. Evaluating this in the context of different $B_0$ and temperatures, we obtain the dependence shown in Extended Data Fig. 3c,d. At ultralow $B_0$, for which a reduction in the control and readout fidelity is seen, $N_{\text{iteration}}$ decreases, possibly because the system deviates from the parity basis. Higher $B_0$ provides a larger qubit energy, increasing the likelihood of obtaining a $|\downarrow\downarrow\rangle$ state after the load ramp and reducing $N_{\text{iteration}}$. Similarly, $N_{\text{iteration}}$ also increases with higher temperatures. At $B_0$ above 1 T, the onset of excited state-level crossings enhances spin randomization after the load ramp, and thus more $N_{\text{iteration}}$ is required. We expect that $N_{\text{iteration}}$ may be reduced by incorporating corrective control based on measurement[9,68] to accelerate the polarization towards the target state.

## SPAM error analysis with repeated readout
A more comprehensive SPAM error analysis uses machine learning of the increased statistics from multiple measurements. The experimental sequence consists of initialization followed by repeated parity readout that results in a series of binary measurement outcomes $m_1, m_2, \ldots, m_n$, where $m_i \in \{\text{evenparity} = 0, \text{oddparity} = 1\}$. This initialization-(measurement)$^n$ sequence is performed 1,000 shots.

A hidden Markov model (HMM) can describe this series of measurements formalism in which the true, but hidden, spin state $s_1, s_2, \ldots, s_n$ follows the Markov chain and the measurement outcomes, $m_i$, are probabilistically related to the underlying spin state. Three different tensors completely determine HMMs:
1. A start probability vector, $\mathbf{\Pi}$, encoding the initializing probabilities in each spin state.
2. A transition probability matrix, $A$, encoding the probabilities of transiting between spin states during measurements.
3. A measurement probability matrix, $\Theta$, encoding the probability of the measurement outcomes conditioned on the current hidden spin state.

To find the likely HMM model for a given set of data, we perform expectation maximization in which we maximize the marginal likelihood, which is dependent on the marginalized hidden spin state, such that

$$L(\mathbf{\Pi}, A, \Theta; \mathbf{m}) := p(\mathbf{m}|\mathbf{\Pi}, A, \Theta)$$
$$= \int p(\mathbf{s}, \mathbf{m}|\mathbf{\Pi}, A, \Theta)\mathrm{d}\mathbf{s}. \qquad (1)$$

For HMM models, there exists the Baum–Welch algorithm that can perform this expectation maximization by an iterative update rule, without the need for backpropagation of gradients[69]. We use the Cramer–Rao bound to quantify the level of uncertainty in these parameters when fitted by expectation maximization[70]. The Cramer–Rao bound states that if $\text{est}_\theta(\mathbf{m})$ is an unbiased estimate of

the parameters $\boldsymbol{\theta} := (\mathbf{\Pi}, A, \Theta)$ given the data $\mathbf{m}$, such as that produced by expectation maximization, then

$$\text{cov}_\theta(\text{est}_\theta(\mathbf{m})) \geq I(\boldsymbol{\theta}; \mathbf{m})^{-1}, \qquad (2)$$

where $I(\boldsymbol{\theta}; \mathbf{y})_{ij} = -\partial^2 \log L(\boldsymbol{\theta}; \mathbf{m})/\partial\theta_i\partial\theta_j$, the Fisher information matrix. Therefore, we can obtain lower bounds on the uncertainty of each parameter from the diagonal elements of the inverse of the Fisher information matrix. We used the Forward–Backward algorithm to compute the marginal likelihood defined in equation (1) needed to compute the Fisher information matrix.

Finally, we use the Viterbi algorithm to compute the most likely set of true spin states that gave rise to the set of measurements given a set of model parameters[69,71].

## Crosstalk correction
The relatively small $\Delta E_Z$ even at higher $B_0$ requires cancellation of crosstalk between the two qubits, that is, the effect on the other qubit when one qubit is being driven. This can be addressed to the first order by considering the following aspects.

To cancel off-resonance driving, we enforce

$$\sqrt{\Delta E_Z^2 + f_{\text{Rabi}}^2} = Nf_{\text{Rabi}}, \qquad (3)$$

where $f_{\text{Rabi}}$ is the Rabi frequency of the target qubit, and $N = 4, 8, 12, \ldots$. Consequently, each π/2 microwave pulse on the target qubit incurs a full $2\pi N$ off-resonance rotation on the ancilla qubit, as exemplified in Extended Data Fig. 7a. Failure to cancel the off-resonance driving can result in large errors under parity readout, as shown in Extended Data Fig. 7b. With $N = 4$, this cancellation criterion dictates the fastest Rabi possible and is therefore expected to limit the single-qubit gate fidelities, especially at low $B_0$ where $\Delta E_Z$ is small. The full set of $f_{\text{Rabi}}$ used for single-qubit randomized benchmarking at different $B_0$ is shown in Extended Data Fig. 7c. In this case, we can alternatively execute X(π/2) as a 3π/2 gate for faster driving at the cost of redundancy. We implemented this with the three- and five-electron qubit at 0.1 T, 1.2 K in Fig. 3d.

In two-qubit sequence runs, it is also necessary to correct AC Stark shift by an amount of

$$\frac{f_{\text{Rabi}}^2}{2\Delta E_Z}, \qquad (4)$$

apart from cancelling the off-resonance driving. Extended Data Fig. 7d measures the AC Stark shift on an ancilla qubit by preparing it on the equator, driving it off-resonantly and projecting the phase. Before the correction, the AC Stark shift is seen as the linear fringes that correspond to the phase accumulation given by equation (4).

We note that the above cancellation of crosstalk does not prevent it from incurring errors. The perturbation on the ancilla qubit induces decoherence. At ultra-low $B_0$ at which $\Delta E_Z$ becomes diminishing, higher-order crosstalk terms cannot be neglected, and the control of individual qubits becomes unmanageable. However, these problems are circumvented in the SMART control scheme, which addresses all the qubits simultaneously.

## Randomized benchmarking
Single-qubit randomized benchmarking sequences for Fig. 3d–e are constructed from elementary π/2 gates [X(π/2), Z(π/2), −X(π/2), −Z(π/2)], π gates [X(π), Z(π)] and an I gate. Each Clifford gate contains one physical elementary gate on average, excluding the virtual Z(π/2) and Z(π) gates.

To optimize the single-qubit gate fidelity, we study different $B_0$ (Fig. 3d) and tightly confine the qubits with low barrier gate voltages to reduce noise coupling. In single-qubit randomized benchmarking, the

coherent driving decouples the qubit from noise to a certain extent[72], and the random rotations of the qubit also have the effect of refocusing[38,73]. Here we optimize the microwave power and thus $f_{Rabi}$, such that the spins are driven quickly without excessive microwave-induced noise[46].

Two-qubit randomized benchmarking sequences for Fig. 4b are constructed from single-qubit elementary $\pi/2$ gates [$X_1(\pi/2)$, $Z_1(\pi/2)$, $X_2(\pi/2)$, $Z_2(\pi/2)$] for Q1 and Q2, and a two-qubit elementary gate DCZ. Each Clifford gate contains 1.8 single-qubit elementary gates and 1.5 two-qubit elementary gates on average. All gates are sequentially executed, which means Q1 idles while $X_2(\pi/2)$ or $Z_2(\pi/2)$ takes place, and the same for Q2. The generated random sequences are used in both randomized benchmarking and FBT. In the case of IRB, we incorporate an interleaved DCZ gate between adjacent Clifford gates. The experimental implementation and the analysis protocol are shown in Extended Data Fig. 9a,b, and the IRB results are shown in Extended Data Fig. 9c.

We then fit the randomized benchmarking decay curve to the formula[38,41]

$$a e^{-(bx)^c} + d, \tag{5}$$

from which $1 - 0.5b$ gives the Clifford fidelity in single-qubit randomized benchmarking, and $1 - 0.75b$ gives the Clifford fidelity in two-qubit randomized benchmarking. The term $c$ represents the decay exponent and reflects the error Markovianity; $a$ is subjected to the readout fidelity and $d$ is close to 0.5.

From the two-qubit IRB decays, we first obtain an IRB fidelity[50] of $99.8 \pm 0.2\%$ at $T = 0.1$ K and $97.7 \pm 1.5\%$ at $T = 1$ K for the DCZ gate. This fidelity reflects the combined effect of dephasing during $t_{exchange}$ and echoing in the DCZ gate and the results can be understood from the stronger temperature dependence of $T_2^{Hahn}$ compared with that of $T_2^*$. We also note the numerical instabilities in IRB fidelities, which result in large error bars.

### Fast Bayesian tomography

FBT[53] is an agile gate set process tomography protocol that can self-consistently reconstruct all gate set process matrices based on previous calibration. In principle, FBT learns and updates the model using the gate sequence information and its experimental outcome. In this work, we feed FBT with the variable-length two-qubit randomized benchmarking sequences and the corresponding experimental data. Clifford gates in the randomized benchmarking sequences are decomposed into their elementary gate implementation of $X_1(\pi/2)$, $Z_1(\pi/2)$, $X_2(\pi/2)$, $Z_2(\pi/2)$ and DCZ. The randomized benchmarking experiments at $T = 0.1$ K and $T = 1$ K run through 32,000 and 26,000 sequences, respectively, sufficient for FBT to reliably reconstruct the error channels. We feed the native parity readout results directly to FBT, without converting them to the standard two-qubit measurement basis.

To initiate the FBT analysis, we must bootstrap the model from educated guesses to help the analysis converge with a finite amount of experiments. Here, we do this by injecting guessed fidelity numbers as introduced in refs. 53,74. FBT models each noisy gate $\widetilde{G}$ as the product of the noise channel $\widetilde{G} = \Lambda G$ and the ideal gate $G$, in which the noise channel $\Lambda$ is linearized about $I$ by expressing it as $\Lambda = I + \varepsilon$. Each update of the FBT analysis is essentially on the statistics of the noise channel residuals $\varepsilon$. Extended Data Fig. 9d shows the reconstructed Pauli transfer matrices of the DCZ gate. Supplementary Information shows the reconstructed noise channel residuals of the three physical elementary gates DCZ, $X_1(\pi/2)$, and $X_2(\pi/2)$ at $T = 0.1$ K and $T = 1$ K.

As FBT does not guarantee that the reconstructed channels are physical or flag any gauge ambiguity, we perform CPTP projection and gauge optimization over the entire gate set at the output stage.

FBT extracts DCZ fidelities of $99.15 \pm 0.13\%$ at $T = 0.1$ K and $98.92 \pm 0.67\%$ at $T = 1$ K. Here, a single-qubit gate on one qubit always leaves the other qubit idling, which considerably limits the single-qubit process fidelities (Supplementary Information) and consequently the Clifford fidelity in two-qubit randomized benchmarking, even at $T = 0.1$ K (Fig. 4b). However, the reduction in the Clifford fidelity from 0.1 K to 1 K mainly originates from the degradation of the DCZ gate, exhibiting a similar factor.

### Error taxonomy with pyGSTi

When examining the fidelity results, we are also interested in understanding the dominant error sources behind the DCZ gate infidelity and their variation at different temperatures. FBT is a flexible and efficient gate set process tomography that enables us to extract gate errors from randomized sequence runs[53,74]. To categorize the gate errors, we perform post-processing of the tomography results obtained by FBT using tools for decomposing errors implemented in the pyGSTi package[59,60].

Error taxonomy for FBT can be achieved by converting the noise channels $\Lambda$ for each gate to their error generator $\mathbb{L}$ using the following relationship:

$$G = \Lambda G_0 = e^{\mathbb{L}} G_0, \tag{6}$$

where $G$ is the estimated noisy gate, and $G_0$ is the ideal gate.

Using the pyGSTi package[59,60], we project $\mathbb{L}$ into the subspace of Hamiltonian and stochastic errors, extracting the coefficients of each elementary error generator. We perform this analysis on each of the gates [DCZ, $X_1(\pi/2)$ and $X_2(\pi/2)$] for both temperatures of 0.1 K and 1 K. The coefficients of the elementary error generators are represented in the Pauli basis and presented in the Supplementary Information. The five largest components of the Hamiltonian and stochastic errors for the DCZ gate are shown in Fig. 4c.

We also estimate the generator or entanglement infidelity $1 - \mathcal{F}_{ent}$ based on these error coefficients, given by[60]

$$1 - \mathcal{F}_{ent} \approx \sum_P s_P + \sum_P h_P^2, \tag{7}$$

where the sum is performed over the extracted coefficients and $P$ denotes non-identity Pauli elements. The approximation is validated by the domination of Hamiltonian errors over stochastic errors in magnitude. To obtain the average gate fidelities ($\mathcal{F}_{avg}$), which are the quantities quoted based on IRB and FBT measurements, it can be connected to $\mathcal{F}_{ent}$ in the following way[75]:

$$\mathcal{F}_{avg} = \frac{d \cdot \mathcal{F}_{ent} + 1}{d + 1}, \tag{8}$$

where $d$ is the dimension of the Hilbert space (4 for a two-qubit system). This means that generally stochastic errors contribute more to the gate infidelities, even in the case in which the magnitudes of the Hamiltonian errors are larger.

### Data availability

The data supporting this work are available at *Zenodo* (https://doi.org/10.5281/zenodo.10452860)[76].

### Code availability

SPAM analysis with machine learning was performed with code from ref. 77. Error taxonomy with pyGSTi was performed with code from ref. 59. The FBT algorithm is given in ref. 53. All other supporting calculations and algorithms are provided in the paper in the form of expressions and diagrams.

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

**Acknowledgements** We acknowledge technical support from A. Dickie. We acknowledge technical discussions on two-qubit initialization with W. Huang, and GST with C. Ostrove and R. Blume-Kohout. We acknowledge support from the Australian Research Council (FL190100167 and CE170100012), the US Army Research Office (W911NF-23-10092), the US Air Force Office of Scientific Research (FA2386-22-1-4070) and the NSW Node of the Australian National Fabrication Facility. The views and conclusions contained in this document are those of the authors and should not be interpreted as representing the official policies, either expressed or implied, of the Army Research Office, the US Air Force or the US government. The US government is authorized to reproduce and distribute reprints for government purposes notwithstanding any copyright notation herein. B.v.S., B.S. and N.A. acknowledge support from the Royal Society (URF-R1-191150) and the European Research Council (grant agreement 948932). J.Y.H., R.Y.S., M.F., S.S., J.D.C., I.H. and A.E.S. acknowledge support from the Sydney Quantum Academy.

**Author contributions** J.Y.H., R.Y.S., A.S., A.L., A.S.D. and C.H.Y. designed the experiments; J.Y.H. performed the experiments under the supervision of A.S., A.L., A.S.D. and C.H.Y.; W.H.L. and F.E.H. fabricated the device under A.S.D.'s supervision on enriched $^{28}$Si wafers supplied by N.V.A., H.-J.P. and M.L.W.T.; S.S. designed the RFSET setup; W.G., N.D.S., S.S., E.V. and A.L. contributed to the experimental hardware setup; W.G., N.D.S. and S.S. contributed to the experimental software setup; J.Y.H., A.S. and C.H.Y. designed the algorithmic initialization protocol with input from R.C.C.L.; B.v.S. and B.S. performed the SPAM error analysis with machine learning under the supervision of A.S. and N.A.; R.Y.S. performed the noise spectroscopy analysis; I.H., A.E.S. and C.H.Y. assisted with the SMART protocol implementation; T.T. assisted with the two-qubit randomized sequence generation; R.Y.S. performed the FBT analysis under the supervision of T.T., A.S. and S.D.B.; M.F. performed the subsequent error generator analysis with pyGSTi under A.S.'s supervision; R.Y.S., W.H.L., M.F., W.G., N.D.S., T.T., J.D.C., C.C.E., S.D.B., A.M., A.S., A.L., A.S.D. and C.H.Y. contributed to the discussion, interpretation and presentation of the results; and J.Y.H., R.Y.S., M.F., B.v.S., F.E.H., S.D.B., A.L., A.S.D. and C.H.Y. wrote the paper, with input from all co-authors.

**Funding** Open access funding provided through UNSW Library.

**Competing interests** A.S.D. is the CEO and a director of Diraq. W.H.L., W.G., N.D.S., T.T., E.V., C.C.E., F.E.H., A.S., A.L., A.S.D. and C.H.Y. declare equity interest in Diraq. J.Y.H., A.S. and C.H.Y. are inventors on a patent related to this work (AU provisional application 2023902138) filed by Diraq with a priority date of 3 July 2023.

**Additional information**
**Correspondence and requests for materials** should be addressed to Jonathan Y. Huang, Andrew S. Dzurak or Chih Hwan Yang.

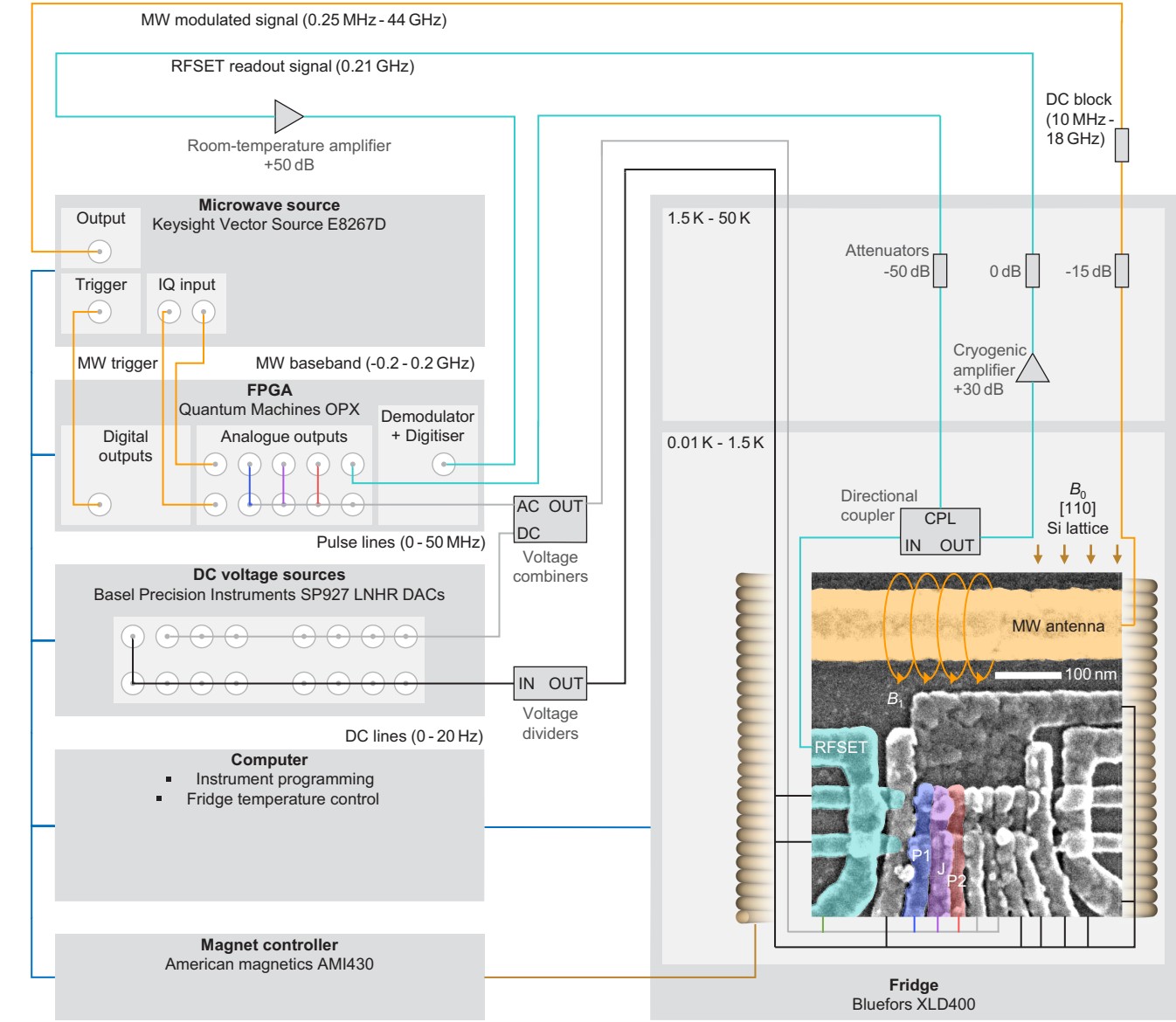

**Extended Data Fig. 1 | Full experimental setup.** Schematic of the measurement setup. See Methods for hardware information.

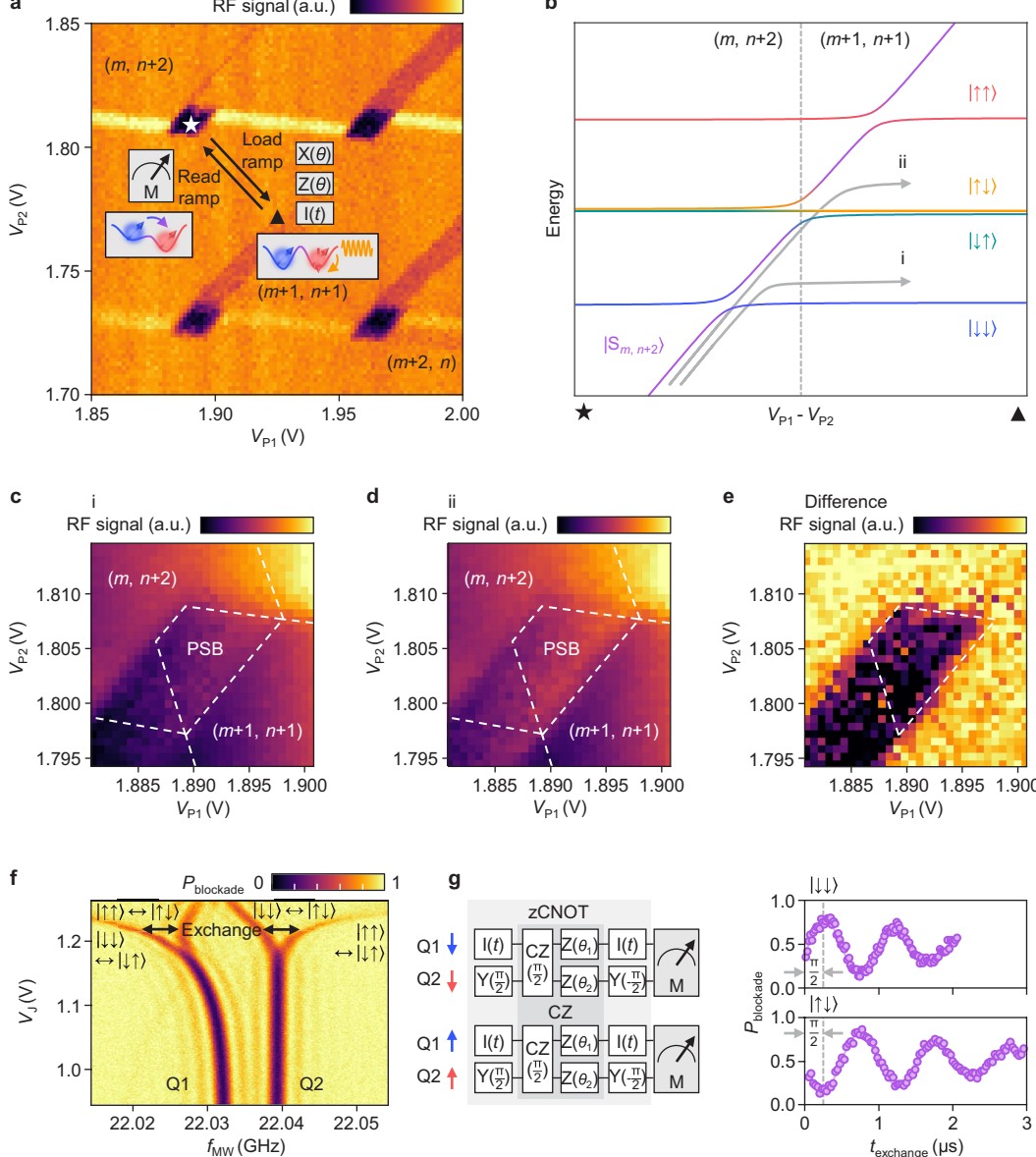

**Extended Data Fig. 2 | Device tune-up. a**, Charge stability diagram as a function of $V_{P1}$ and $V_{P2}$ showing the operation regime. The readout and control point are labelled with star (★) and triangle (▲). **b**, Schematic energy diagram of a double-dot system across the inter-dot transition between the $(m, n + 2)$ and $(m + 1, n + 1)$ charge state, where $m$ and $n$ are even numbers. $(m + 1, n + 1)$ represents a charge state with an unpaired electron spin in each of the dots. The two arrows labelled i and ii refer to two possible loading mechanisms, i being the more adiabatic process. **c**, Averaged signal from 50 shots of charge readout around the $(m + 1, n + 1)$-$(m, n + 2)$ transition, showing partial blockade. The electrons are initialised into $(m + 1, n + 1)$ via an adiabatic ramp which tends to incur a lowest-energy $|\downarrow\downarrow\rangle$ state. **d**, Averaged signal from 50 shots of charge readout around the $(m + 1, n + 1)$-$(m, n + 2)$ transition, with the $(m + 1, n + 1)$ state diabatically

initialised, which can result in the odd-parity states or even $|\uparrow\uparrow\rangle$. Consequently, the blockade is fainter. **e**, Difference between the readout signals in c and d. At millikelvin temperatures, it is possible for this bias in the spin proportions to be large enough for high-fidelity initialisation. The bias is reduced with the increased thermalisation above 1 K, but nonetheless visible here when comparing the resulting PSB from two vastly different initialisation ramp rates. **f**, ESR spectrum as a function of $V_J$, showing the exchange opening up at $V_J$ above 1.1 V. **g**, Construction of a zero-CNOT (zCNOT) gate[23] and calibration of the encompassed CZ gate. The CZ gate consists of a CZ operation with $\pi/2$ duration, followed by single-qubit virtual phase corrections to account for the Stark shifts.

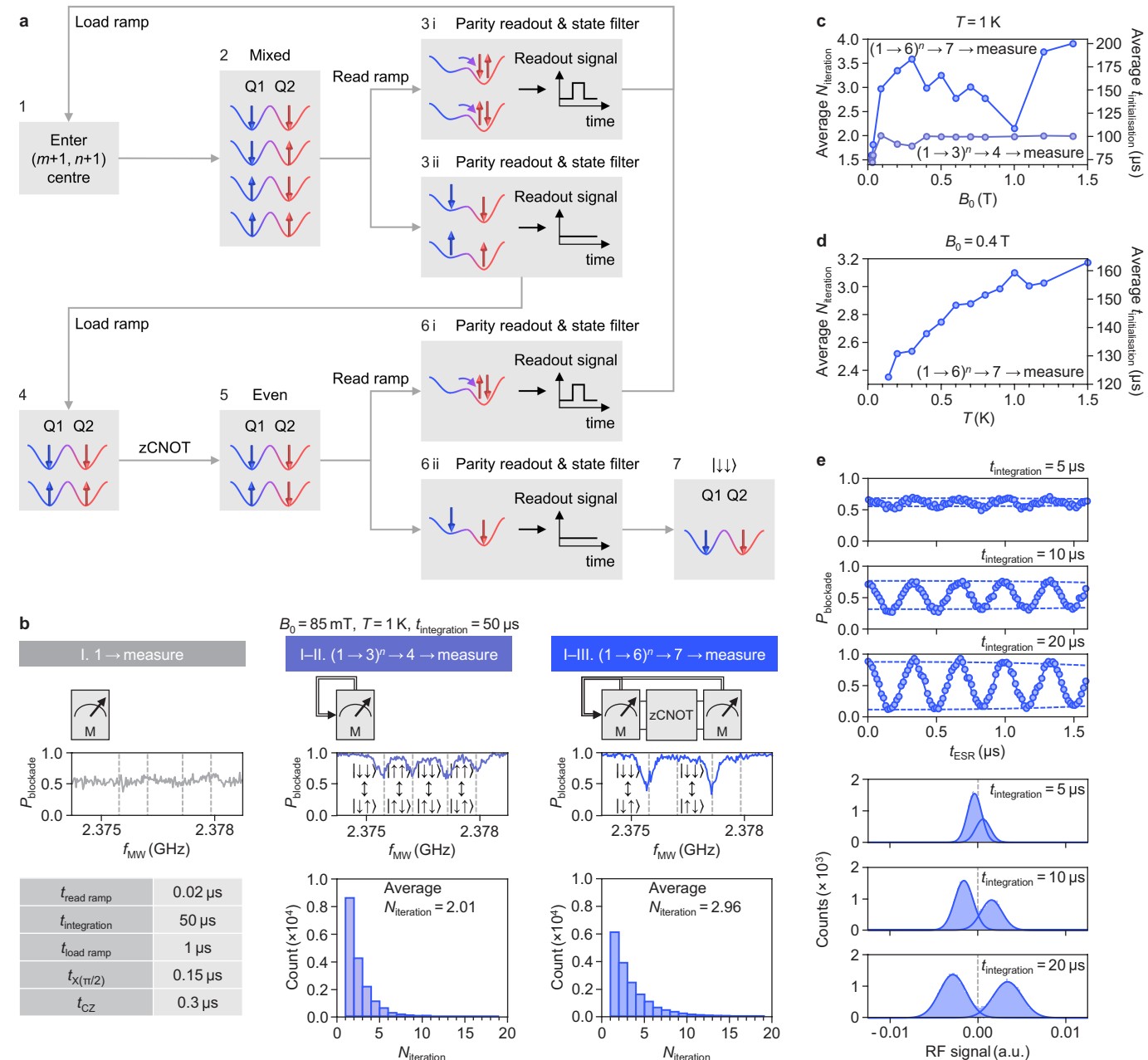

**Extended Data Fig. 3 | Two-qubit algorithmic initialisation. a**, Full protocol of the algorithmic initialisation of a two-qubit system, based on parity readout. **b**, Experiments with various stages of the algorithmic initialisation and the corresponding ESR spectra. The data are taken at $T = 1\,K$ and $B_0 = 85\,mT$, where the thermal energy is 8 times greater than the qubit energies. The table shows the nominal duration for each part of the operation used in this work. Initialisation with only Step 1 corresponds to the conventional ramped initialisation. The first part of the algorithmic initialisation repeats Step 1 to 3 until an even-parity state is detected. The full algorithmic initialisation repeats Step 1 to 6 in order to detect if the state is solely $|\downarrow\downarrow\rangle$. With the partial or the full

algorithmic initialisation, measured with 20000 shots each, we record the statistics on the numbers of iterations, $N_{iteration}$, and evaluate the respective average $N_{iteration}$. **c**, Average $N_{iteration}$ as a function of $B_0$ at $T = 1\,K$. Taking the duration amounts from b into account, the average time cost for initialisation, $t_{initialisation}$, is estimated. **d**, The quantities in c as a function of temperature at $B_0 = 0.4\,T$. **e**, Rabi oscillations and charge readout histograms with different amounts of readout integration time, $t_{integration}$, at $B_0 = 0.4\,T$, $T = 1\,K$. With short $t_{integration}$, the Rabi amplitude becomes limited by the charge readout instead. This may be improved by more advanced readout techniques, such as a double-island SET[78] operating at RF or gate dispersive readout[79].

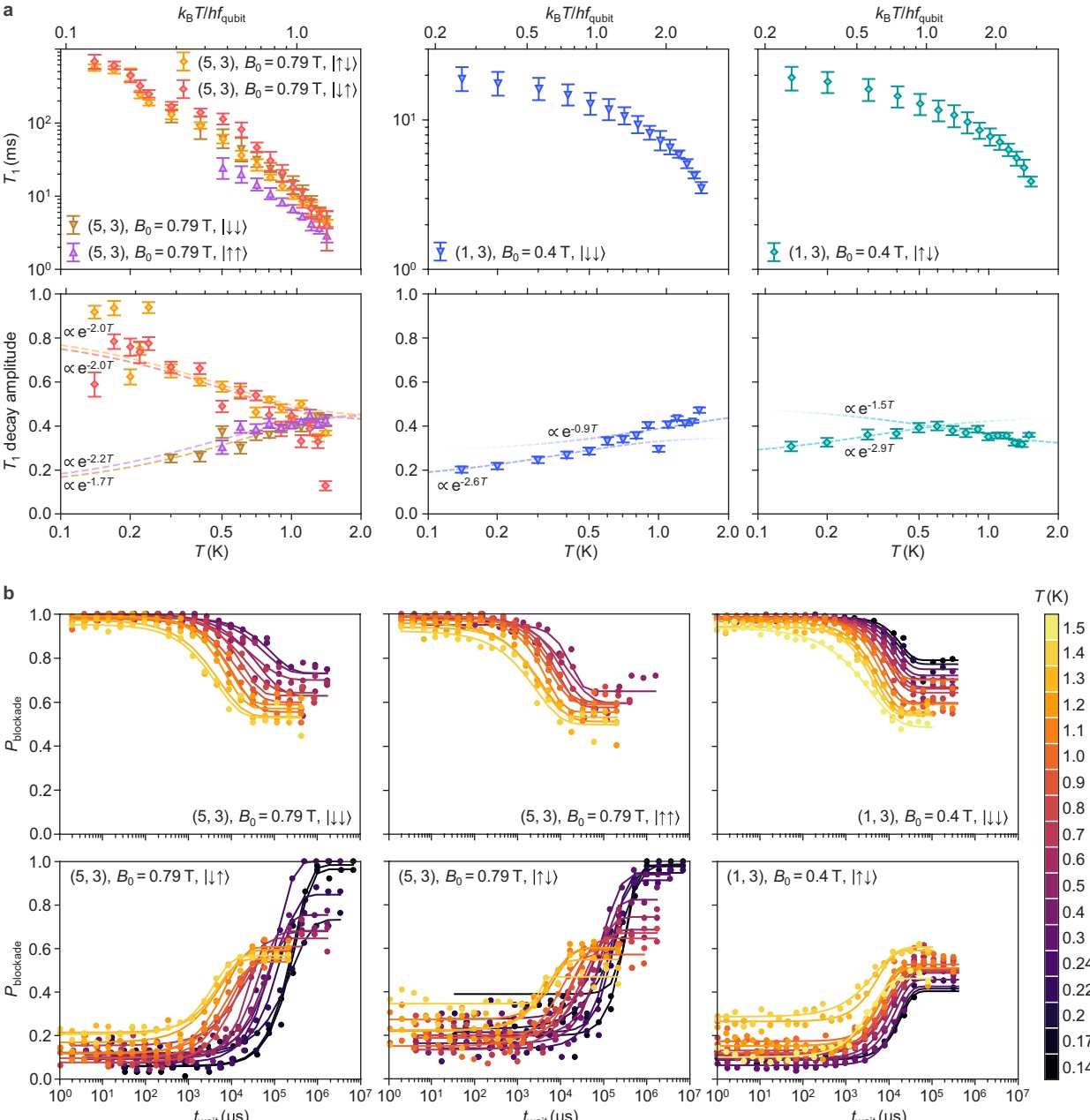

**Extended Data Fig. 4 | $T_1$ processes and temperature dependence. a,** The characteristic time of spin relaxation, $T_1$, and the decay amplitude for various two-qubit states as a function of temperature. We recognise the presence of different relaxation mechanisms at low temperatures, as described in the main text. Here we also look at the evolution of the decay amplitudes, defined as the difference in $P_{blockade}$ between the starting point and decay equilibrium. At low temperatures where the relaxation to low-energy states dominates, the decay reaches an equilibrium with mostly $|\downarrow\downarrow\rangle$. With even-parity initialisation, the decay amplitude should be well below 0.5. With odd-parity initialisation, the decay amplitude should be well above 0.5. At high temperatures where the thermal energy becomes comparable or greater than the qubit energy, the decay equilibrium is a mixed state and $P_{blockade}$ tends towards 0.5. Therefore, the decay amplitude reduces as the temperature increases, following an $e^{-k_B T}$-like

reduction, until the degradation of readout starts to dominate. This trend is apparent in the (5, 3) state, but becomes more convoluted in (1, 3), possibly due to lower-lying excited states. Although $T_1$ is not the limiting time scale in this temperature range, we recognise the rich physical processes behind relaxation revealed in this work additional to the previous results[15,16] and their potential impact on longer or higher-temperature operation in the future. **b,** Measured and fitted relaxation decay curves. Since all the decay curves are one-way, they are fit to a single formula $ae^{-(t/T_1)^c} + d$, where $a$, $c$ and $d$ are the decay amplitude, exponent and equilibrium. Although fluctuations in the readout level is inevitable before RFSET feedback takes place at the end of each shot, the two-level separation in the charge readout is sufficiently large to maintain an overall correct readout level (Supplementary Information). Error bars represent the 95 % confidence level.

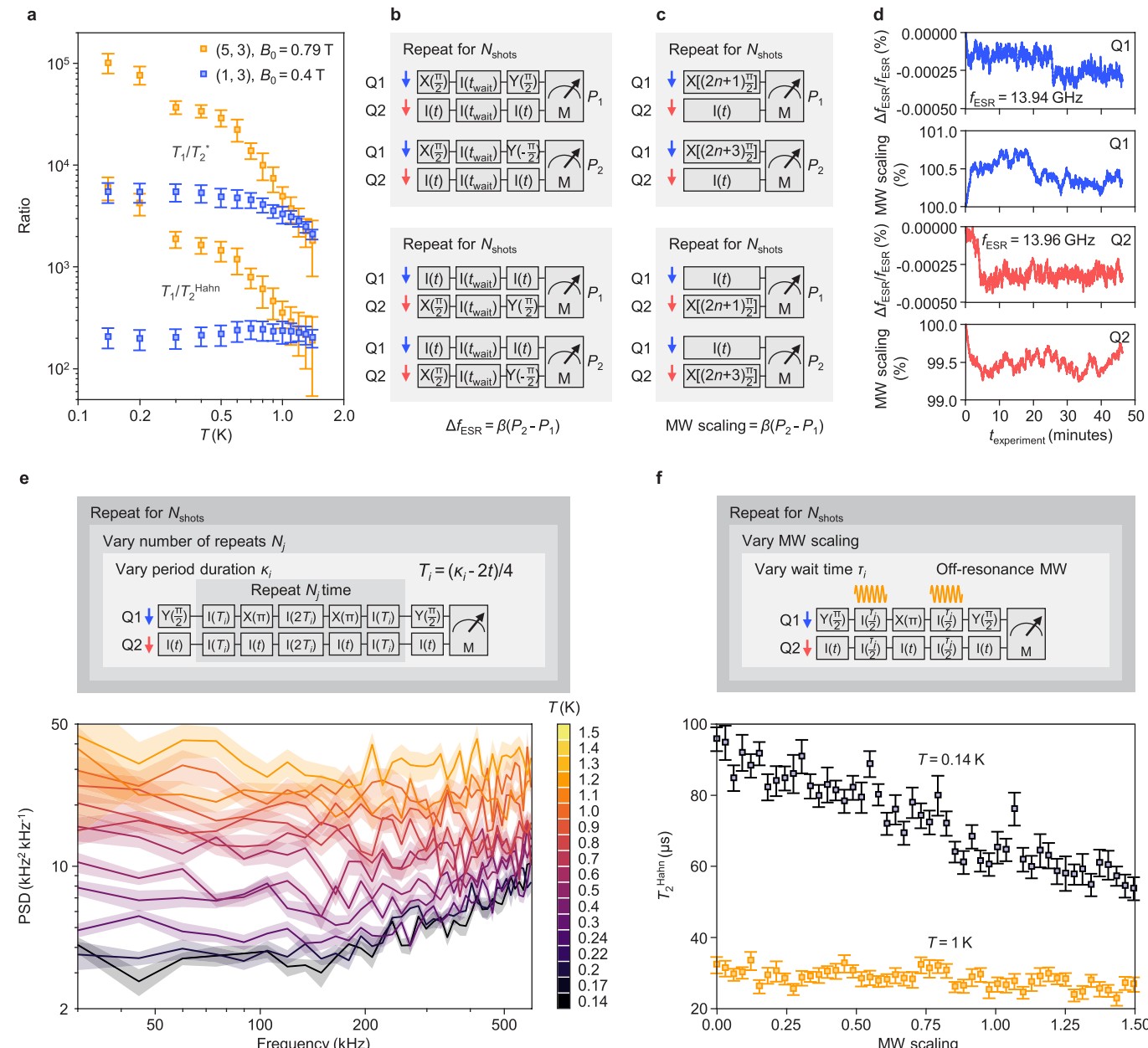

**Extended Data Fig. 5 | Single-qubit temperature dependence, stability, and noise characteristics. a**, Ratio of $T_1$ to $T_2$ as a function of temperature in different regimes. This ratio indicates the amount of bias in the proportion of depolarisation errors to that of dephasing errors. A large variation in the bias and its temperature dependence is seen at temperatures below 1 K, whereas these metrics become similar above $T = 1$ K. At this point, $T_1/T_2^*$ shows a high-order roll-off. However, the temperature dependence is weaker when echoing is incorporated, as seen in $T_1/T_2^{\mathrm{Hahn}}$. The overall $T_1/T_2$ biases remain above 100 within $T = 1.5$ K. **b**, Sequences for tracking slow changes in $f_{\mathrm{ESR}}$ over a long time with respect to $T_2^{49}$. **c**, Sequences for tracking the amount of adjustment in microwave power to maintain a constant $f_{\mathrm{Rabi}}$ over time[48]. $P_1$, $P_2$

correspond to the different projection outcomes and $\beta$ is a conversion factor. **d**, Results of a and b at $B_0 = 0.5$ T and $T = 1$ K. $P_1$, $P_2$ correspond to the different projection outcomes and $\beta$ is a conversion factor. **e**, Sequence for the noise spectroscopy based on the Carr-Purcell-Meiboom-Gill (CPMG) protocol[44,45] and the full set of noise spectra of Q1 at temperatures from 0.14 K to 1.2 K. **f**, We examine the microwave effect on the qubit coherence time by applying the Hahn echo sequence on Q1. During the wait time, we apply a microwave signal far from the resonance of either qubit to capture the incoherent noise induced. We measure $T_2^{\mathrm{Hahn}}$ varying the microwave power at $T = 0.14$ K and $T = 1$ K. We observe a notably less evident effect from the microwave at $T = 1$ K compared to at $T = 0.14$ K. Error bars represent the 95 % confidence level.

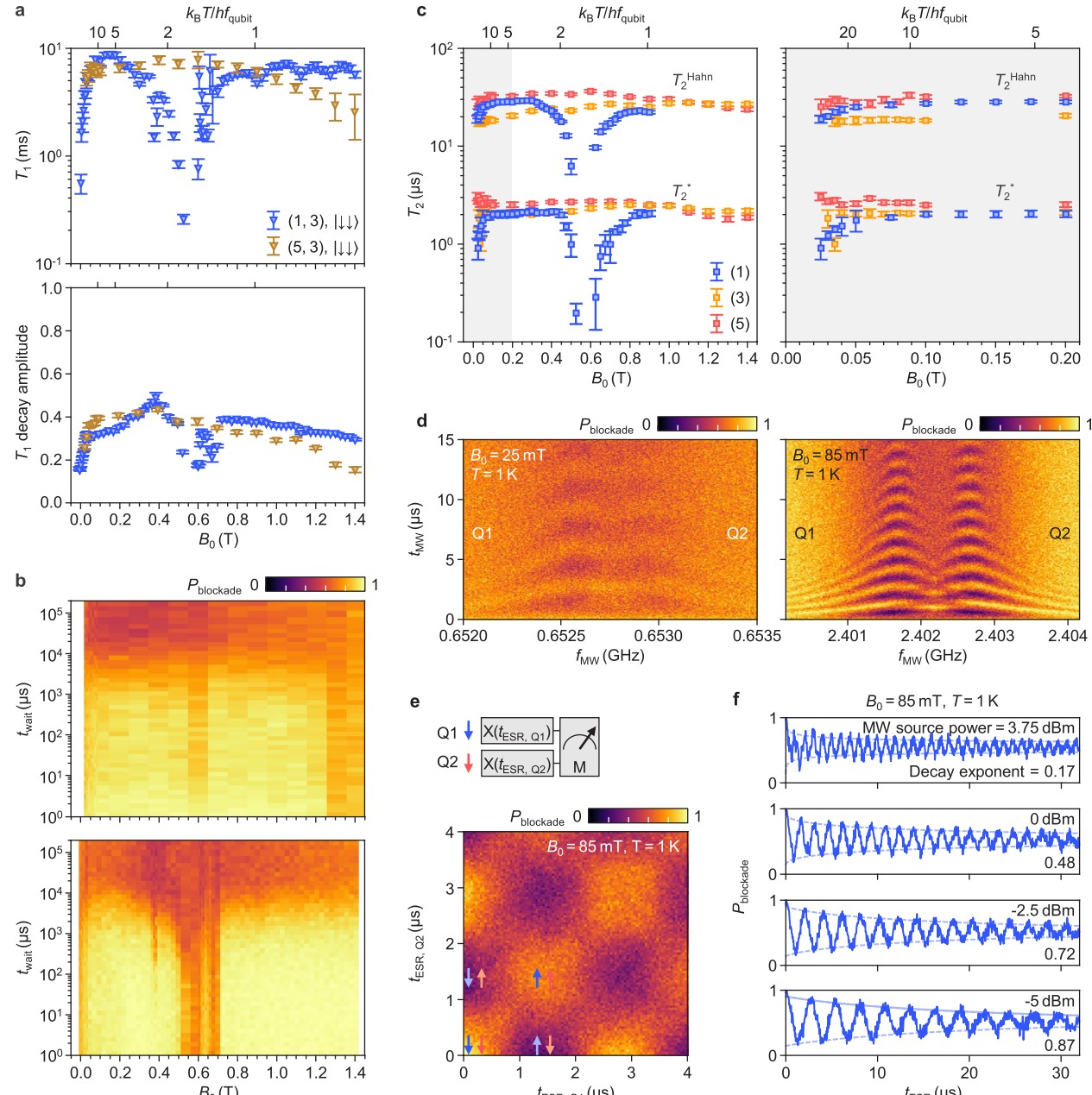

**Extended Data Fig. 6 | $B_0$ dependence. a**, $T_1$ and the decay amplitude as a function of $B_0$ at $T = 1$ K. In (1, 3), $T_1$ exhibits a notable drop at low $B_0$ and near the hot spot induced by excited state crossings. The reduced decay amplitude is caused by the degraded spin readout around the hot spot, and additionally the small qubit energy relatively to the thermal energy at low $B_0$. **b**, Measured $T_1$ decay curves as a function of $B_0$ at $T = 1$ K. The curves are fitted with the same method as described in Extended Data Fig. 4b. **c**, $T_2$ as a function of $B_0$ down to 25 mT at $T = 1$ K in several charge configurations. $T_2^*$ and $T_2^{Hahn}$ are almost $B_0$-invariant in the three- and five-electron configurations, but experience a drop around the hot spot in the one-electron configuration. The effect is highly local and the qubit performance is consistent across configurations at low $B_0$ until 50 mT. **d**, Rabi oscillations in (5, 3) at ultra-low $B_0$ of 25 mT and 85 mT, where the qubit energy is only 3.3 % and 11.4 % of the thermal energy.

Due to the small $\Delta E_z$, crosstalk and deviation from the standard parity basis $\{|\downarrow\downarrow\rangle, |\downarrow\uparrow\rangle, |\uparrow\downarrow\rangle, |\uparrow\uparrow\rangle\}$ become severe. **e**, Simultaneously driven Rabi oscillations on both qubits showing the alternation of the four parity basis states. **f**, Resonant Rabi oscillation of Q1 as a function of microwave power at $B_0 = 85$ mT and $T = 1$ K. The decay envelops are fitted to $ae^{-(t/T_2^{Rabi})^c} + d$, where $a$ and $c$ are the decay amplitude and exponent and $d$ is around 0.5. In general, we notice a reduction in the decay exponent at lower $B_0$, especially with faster driving. Possible causes are off-resonance driving on the ancilla qubit, decoherence during off-resonance driving, or a greater effect from the microwave-induced noise[46]. All of these can contribute to the reduced oscillation amplitude. The coherence does not appear to be affected and the quality factor of the Rabi oscillation is improved with faster driving. Error bars represent the 95 % confidence level.

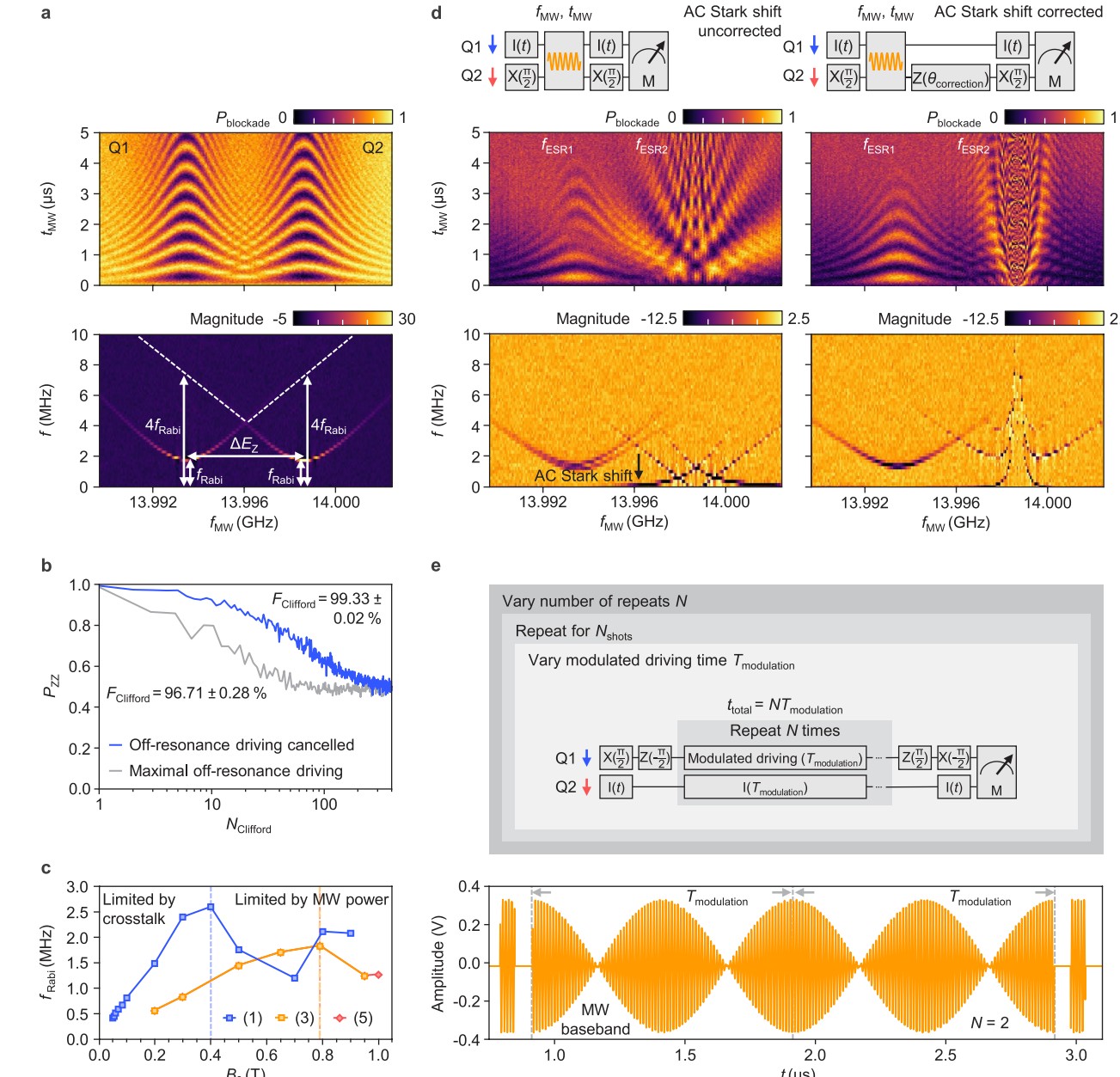

**Extended Data Fig. 7 | Qubit crosstalk and the SMART protocol. a**, Crosstalk due to off-resonance driving at $B_0 = 0.5$ T and $T = 1$ K, plotted in time and frequency domain. In this measurement, $\Delta E_Z$ and microwave power are set such that when Q1 is resonantly driven at $f_{Rabi}$, the Rabi frequency of the off-resonance driving on Q2 is exactly $4f_{Rabi}$ to meet the cancellation condition in equation (3), with $N = 4$. This also applies to the case where Q2 is resonantly driven and Q1 is off-resonantly driven. **b**, Single-qubit randomised benchmarking (RB) of Q1 with and without off-resonance driving at $B_0 = 0.5$ T and $T = 1$ K. We maximise and cancel the off-resonance driving using the relationship in a. **c**, $f_{Rabi}$ used in single-qubit RB at different $B_0$. This is set to meet the off-resonance driving cancellation condition based on the $\Delta E_Z$ in each $B_0$ and charge configuration following equation (3). We use $N = 4$ for fast driving until we reach the limit of

the microwave source at high $B_0$, where the power transmission in the microwave line becomes much weaker. **d**, Sequence for probing the AC Stark shift and the results in time and frequency domain, taken at $B_0 = 0.5$ T and $T = 1$ K. In this example, we use Q2 to probe the AC Stark shift: we prepare it in the -Y direction and apply a microwave pulse with varying frequency $f_{MW}$ and duration $t_{MW}$. We show the results with and without correction. Without correction, AC Stark shift is seen as the linear fringes, which will translate into coherent Z errors during two-qubit operation. **e**, Sequence for the SMART dressing protocol[8]. The sequence prepares the target qubit along the +X axis, and drive it with a cosine-modulated microwave pulse for a duration of $T_{modulation}$. The qubit is then projected back onto the +Z axis for measurement.

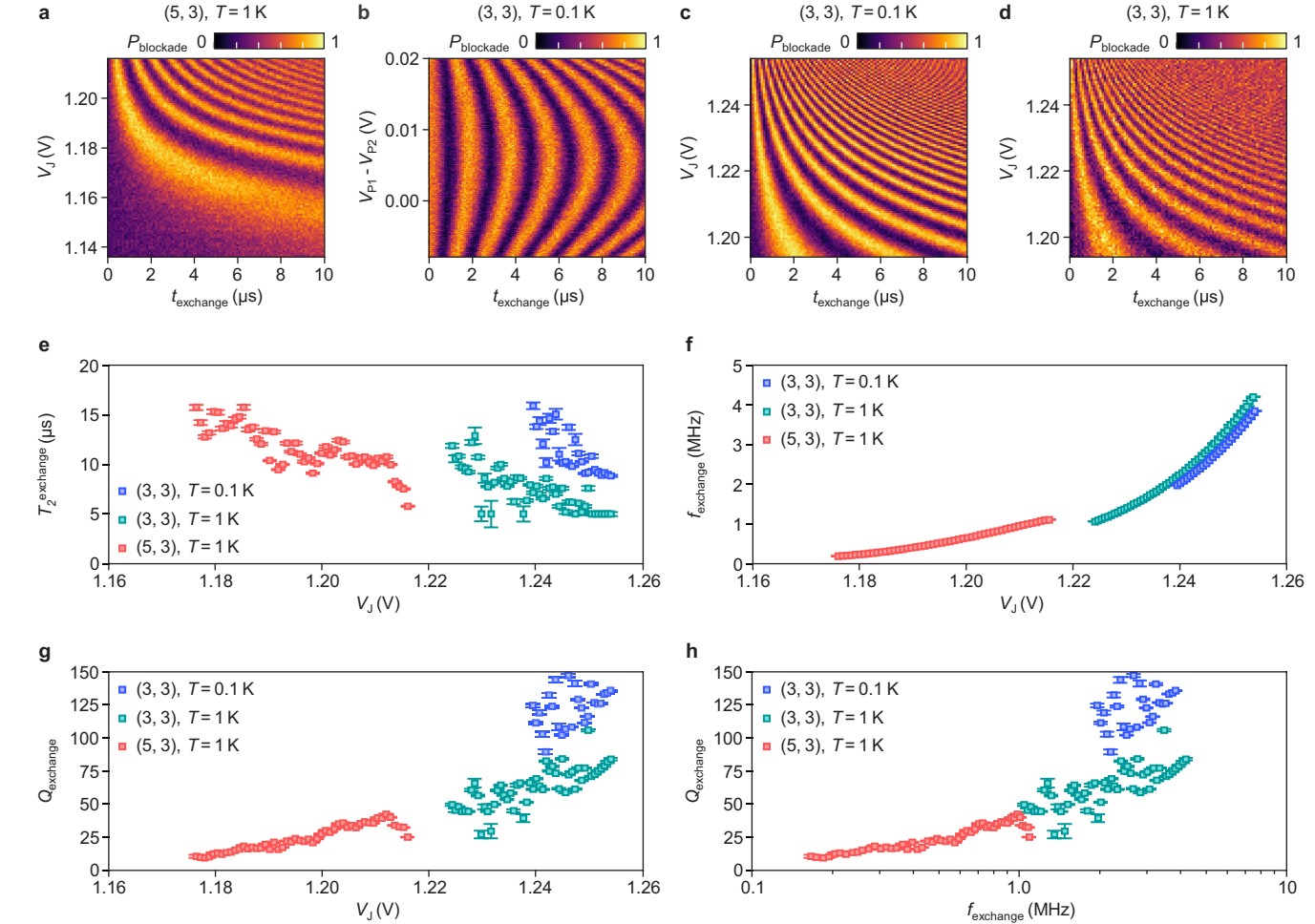

**Extended Data Fig. 8 | Tuning of DCZ oscillations. a**, DCZ oscillations in (5, 3) as a function of time and $V_J$ at $B_0 = 0.79$ T and $T = 1$ K. **b**, DCZ oscillations in (3, 3) as a function of time and $V_{P1} - V_{P2}$ at $B_0 = 0.79$ T and $T = 1$ K showing the symmetric operation point [36,50,80,81]. **c, d**, DCZ oscillations in (3, 3) as a function of time and $V_J$ at $B_0 = 0.79$ T and $T = 0.1$ K and 1 K. **e**, $T_2$ of the DCZ oscillations $T_2^{exchange}$ as a function of $V_J$ at $B_0 = 0.79$ T, $T = 1$ K. **f**, Frequency of the DCZ oscillations $f_{exchange}$ as a function of $V_J$ at $B_0 = 0.79$ T, $T = 1$ K. **g**, Quality factor of the DCZ oscillations $Q_{exchange}$ as a function of $V_J$ at $B_0 = 0.79$ T, $T = 1$ K, indicating that $f_{exchange}$ at higher $V_J$ outpaces $T_2^{exchange}$. **h**, $Q_{exchange}$ as a function of $f_{exchange}$ at $B_0 = 0.79$ T, $T = 1$ K. Error bars represent the 95 % confidence level.

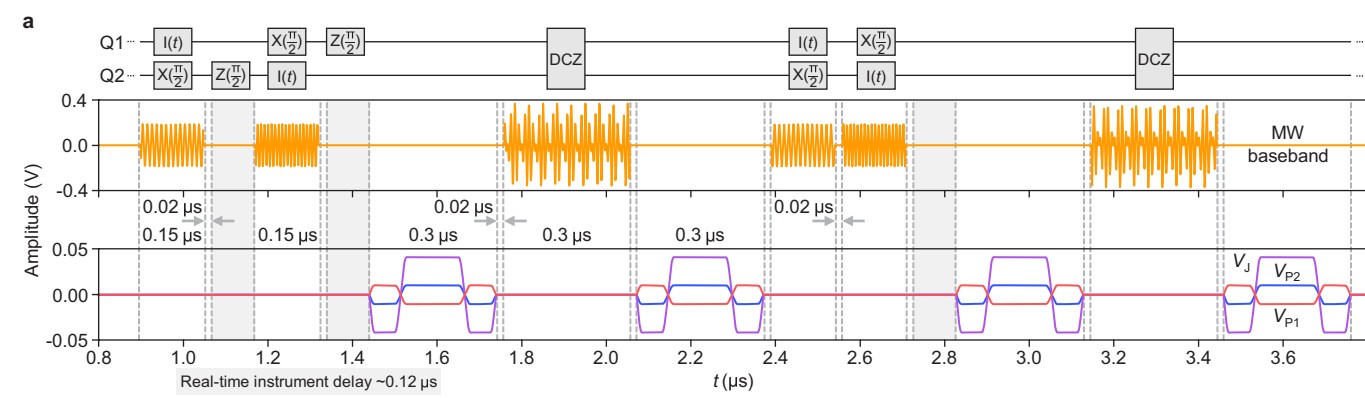

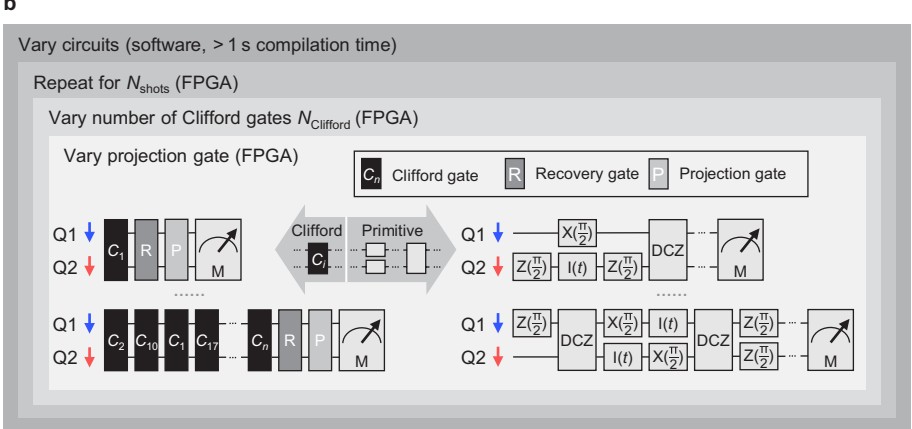

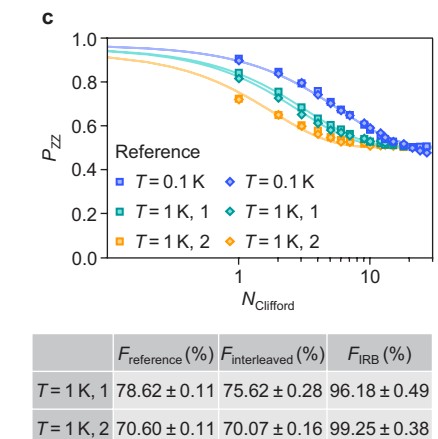

| | $F_{reference}$ (%) | $F_{interleaved}$ (%) | $F_{IRB}$ (%) |
|---|---|---|---|
| $T = 1$ K, 1 | 78.62 ± 0.11 | 75.62 ± 0.28 | 96.18 ± 0.49 |
| $T = 1$ K, 2 | 70.60 ± 0.11 | 70.07 ± 0.16 | 99.25 ± 0.38 |

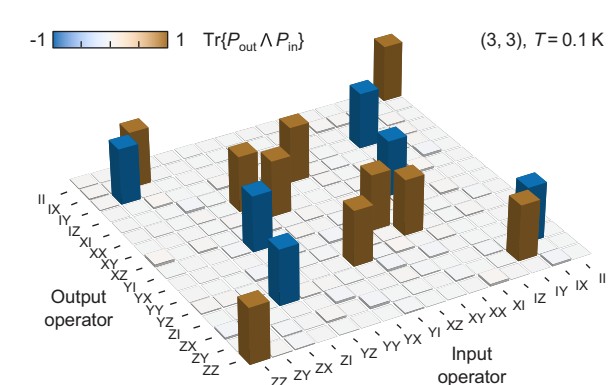

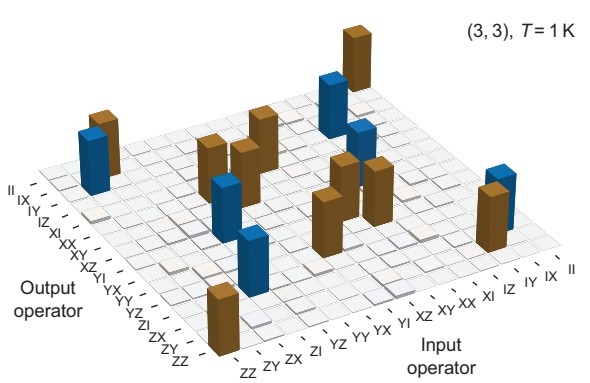

**Extended Data Fig. 9 | Benchmarking and tomography of universal two-qubit logic. a**, An example random sequence with the microwave and voltage pulses generated by the FPGA. A DCZ gate includes two voltage pulses separated by an echoing two-tone X(π) pulse. The voltage pulse shape is designed to cancel any slow drift and compensation is applied to P1, P2 while exchange is being pulsed. We set a padding of 0.02 μs between two adjacent pulses. The real-time logic required for the FPGA to apply the sequences incur unintended gaps in the order of 0.1 μs between some of the quantum gates. This introduces both coherent and incoherent errors. Ensuing efforts should

target the minimisation of real-time logic and accurate synthesis of waveforms prior to the sequence run. **b**, The experiment and analysis protocols for two-qubit randomised benchmarking and FBT. The experimental gate sequences consist of random Clifford gates $C_i$ in the two-qubit space with a recovery gate R at the end. We then perform a projection P in +ZZ (no operation before parity readout) projection and -ZZ (π pulse on a single qubit before parity readout). **c**, IRB results at $B_0 = 0.79$ T, $T = 0.1$ K and 1 K. **d**, Pauli transfer matrices (PTMs) for the DCZ gate at $B_0 = 0.79$ T, $T = 0.1$ K and 1 K, determined by FBT. Error bars represent the 95 % confidence level.

**Extended Data Table 1 | Key metrics of the two-qubit processor**

| Operating condition | | Fidelity (%) | | | | |
|---|---|---|---|---|---|---|
| External magnetic field | Temperature | Initialisation even | Readout even | Readout odd | 1Q Clifford gate | DCZ gate |
| $B_0 = 0.79$ T ($f_{Rabi} = 1.84$ MHz) | $T = 0.1$ K | $99.40 \pm 0.25$ | $99.69 \pm 0.07$ | $96.79 \pm 0.12$ | - | $99.15 \pm 0.13$ |
| | $T = 1$ K | $99.34 \pm 0.27$ | $99.34 \pm 0.08$ | $96.15 \pm 0.44$ | $99.60 \pm 0.01$ | $98.92 \pm 0.67$ |
| $B_0 = 0.4$ T ($f_{Rabi} = 2.6$ MHz) | $T = 0.14$ K | - | - | - | $99.89 \pm 0.01$ | - |
| | $T = 1$ K | - | - | - | $99.85 \pm 0.01$ | - |

| Operating condition | | Relaxation time (ms) | Dephasing time (µs) | | Error bias | |
|---|---|---|---|---|---|---|
| External magnetic field | Temperature | $T_1$ | $T_2^*$ | $T_2^{Hahn}$ | $T_1/T_2^*$ | $T_2^{Hahn}/T_2^*$ |
| $B_0 = 0.79$ T | $T = 0.14$ K | $331.29 \pm 78.00$ | $3.44 \pm 0.13$ | $76.86 \pm 17.08$ | $961305 \pm 262661$ | $22 \pm 6$ |
| | $T = 1$ K | $9.29 \pm 3.99$ | $2.32 \pm 0.19$ | $33.26 \pm 3.38$ | $4004 \pm 2048$ | $14 \pm 3$ |
| $B_0 = 0.4$ T | $T = 0.14$ K | $19.48 \pm 3.64$ | $3.60 \pm 0.14$ | $95.85 \pm 2.41$ | $5411 \pm 1207$ | $27 \pm 2$ |
| | $T = 1$ K | $7.74 \pm 1.20$ | $2.32 \pm 0.10$ | $32.65 \pm 1.32$ | $3336 \pm 661$ | $14 \pm 1$ |

Error bars represent the 95% confidence level.