## [Peer Review File · Nature]

Manuscript Title: High-fidelity spin qubit operation and algorithmic initialisation above 1 K

e 1 K

Reviewer Comments & Author Rebuttals

Reviewer Reports on the Initial Version:

Referees' comments:

Referee #1 (Remarks to the Author):

In this manuscript, the authors present an impressive set of spin qubit results at a temperature of 1K. The main new feature of this work is the improvement of two-qubit gate fidelity at 1K from previously reported values around 90% to a new record around 99%. This is not only important for eventual surface code viability, but it already enables a new algorithmic initialization procedure.

My technical comments are as follows.

1) On line 84 the authors refer to the valley splitting being smaller than the orbital excitation energy. It would be good to include those numbers for this device.

2) Related to the previous comment, why is a (1,3) charge state included in Figure 3? It seems that the authors have already stated that this should not be used due to potential valley excitations. And why is (3,3) not included?

3) The abstract and introduction mention the one-qubit and two-qubit gate fidelities and SPAM fidelities being in the fault-tolerant range, but it seems to me this is slightly (unintentionally) misleading because the quoted fidelities are for different charge regimes. Figure 3 shows good one-qubit gate fidelities with either (1,3) or (5,3), while it appears from Supplementary Figure 6 that fidelities in (3,3) are much worse. Figure 4 gives good two-qubit gate fidelities for (3,3), but Extended Data Figure 8 shows that operating in (5,3) gives significantly worse quality factors. I could not find a specification of the charge regime the SPAM results were for. This should be clarified. In any case, it appears this device cannot achieve the fault-tolerant range in all aspects simultaneously at 1K.

4) In line 313, I don't think it is quite fair to say that RB is not reflecting the "true DCZ fidelity." The nonideal echo is part of the gate. It seems that what the authors have in mind as the "true" infidelity is the part that originates during the exchange pulses, excluding imperfections in the echo, and the FBT result can extract that. While the FBT analysis is of interest for diagnosing how to improve further, what is relevant for operation of a quantum processor is the infidelity of the full DCZ operation, which is here limited to around 98% by the single-qubit idle time during the echo. This is still an impressive improvement! I simply think it is confusing to report the FBT fidelity as the true fidelity, and it should be clarified that the RB fidelity is the more relevant one for fault tolerance.

5) In line 363 the authors mention the necessity of "advanced pulse shaping techniques to eliminate

the non-Markovian noise sources." Some representative references here would be helpful.

6) It would be useful for the authors to say what were the main factors in the improvement of 1K two-qubit fidelity compared to Ref. 16.

In conclusion, this manuscript is a remarkable improvement in the state of the art, with exciting ramifications for the prospects of spin qubit scalability. I would recommend it for publication in Nature provided the comments above are satisfactorily addressed.

Referee #2 (Remarks to the Author):

In "High-fidelity operation and algorithmic initialization of spin qubits above one kelvin", Huang et al. describe a novel protocol for two-qubit state preparation which they use to successfully prepare low entropy spin states that are far out of equilibrium with the local spin bath at 1K. Furthermore, they combine this initialization protocol with single- and two-qubit universal control and spin parity readout, all at 1K, to achieve near state-of-the-art fidelities comparable to recent spin qubit results operating at millikelvin temperatures.

In my opinion, the demonstration of a novel initialization protocol at temperatures approaching the single qubit energies, along with the successful execution of universal control in quantum dots, is a novel and outstanding result that I do think is worthy of publishing at Nature.

However, I do have some lingering issues that I would like the authors to address.

1. In Fig. 1(c), a charge stability diagram is marked with a 'readout' operating point near the $(m, n+2) \rightarrow (m+1, n+1)$ charge transition, located about $VP1-VP2 \sim -0.1V$. But in Fig. 1(d), the author's show a 2D sweep of readout position vs blockade probability, swept over a narrow range about $VP1-VP2 \sim 0V$. Shouldn't the bias sweep of Fig.1(b) be near the bias conditions for the readout position in Fig.1(c)? And if not, I suggest that the author's clarify this point in the caption.

2. The author's state that they "prefer to load three electrons in at least one of the dots to avoid the small excitation energy" (line 82). However, there is no mention of expected valley energies in such configurations and, given that the device is operating at elevated temperatures, there is no consideration for the probability of generating exciting states outside of the spin polarization basis that the author's use to describe their initialization protocol. I would imagine that their protocol should successfully filter out such excited valley states, as such states would not be blockaded by PSB readout and thus would be rejected in stage ii of the initialization protocol. Is this correct? In either case, I suggest that the authors should explicitly address the issue of active initialization in the (potential) presence of excited states. Such comments would not only strengthen the arguments of the paper but also relax any implicit assumptions for scalable high-fidelity initialization.

3. On lines 221 – 223, "in this device, we expect the reduced charge and magnetic noise ...", the authors state that the device used in this demonstration has improved noise performance. As far as I

can tell, there is no discussion or explanation for such improvements. Please address the source(s) of such improvements.

4. On lines 261-265, "... with the strong capability of noise decoupling [...] and fast spin rotations [...], the optimized single-qubit Clifford fidelity ...", the authors imply that there is some amount of noise decoupling taking place in the gates. However, there is no explicit mention of single-qubit gate optimization in Methods F. Have the authors explicitly incorporate any explicit noise mitigation techniques here?

5. Please comment on the inconsistency between the quoted SPAM (readout + initialization) fidelity metrics of ~99%, compared to the full visibility of Rabi oscillation [Extended Data Fig. 6(f)], DCZ oscillations [Fig. 4(a)] and +Z/-Z RB decays (Supplementary Fig. 5), which all look to be <90%. I understand there are many reasons that could explain this difference (different device configurations, tune-ups, or different sequences have predictably different visibility, etc) but an explanation from the author's would be very helpful.

6. Please comment on the use of fidelity "ratios" for reporting IRB fidelities. I am confused if this is referring to (1) the gate error deduced from the difference between interleaved and reference RB sequences, or (2) the actual ratio of sequence decay rates. I am familiar with reporting the interleaved gate fidelity directly, and less familiar with ratios. Perhaps this is my misunderstanding of notation, but if not, please cite any supporting material for the justification of IRB ratios.

Referee #3 (Remarks to the Author):

The manuscript by Huang et al marks a step forward in the strive to increase the operation temperature of silicon spin qubits. Earlier works by the same research group and others had already provided some first proof-of-concept demonstrations showing a moderate loss of qubit performance when raising temperature from tens of mK to about 1K. Huang et al report further progress by introducing a new type of initialization procedure based on the Pauli spin blockade effect in combination with zCNOT conditional rotation. The achieved initialization fidelity exceeds 99% for the even-parity, two-qubit ground state.

The paper is complemented by a thorough characterization of one- and two-qubit gate fidelities, with results surpassing the current state-of-the-art. In particular, the fidelity of two-qubit, decoupled control-phase (DCZ) operations approaches 99%. The authors offer an extensive discussion of the possible causes of infidelity.

Overall, I find this work rich and scientifically sound. On a technical level, I have no serious criticism to raise. On the contrary, I am very impressed by the massive amount of meticulously performed, highly valuable work. The authors deploy a variety of sophisticated techniques, including the use of machine learning for error analysis.

That said, this work is to my view unsuitable for publication in Nature. While it is certainly relevant to the spin qubit community, its novelty level lies well below the typical standards of a journal like Nature. The demonstrated initialization protocol is to a good extent original but I cannot view it as a clear breakthrough: 1) Pauli spin blockade is well known to work at temperatures well above the

qubit characteristic energy (e.g., it was used in the earlier demonstrations of 1K spin-qubit operation by Yang et al. and Petit et al., ie refs 16 and 17, respectively); 2) Moreover, in refs 16 and 17, initialization fidelity was not measured and charge readout was performed by measuring current through a nearby SET, as opposed to the rf-SET readout technique used in this work. Therefore, it is difficult to judge the level of improvement with respect to those earlier works. The one- and two-qubit fidelities measured by Huang et al do exceed the current state-of-the-art for 1K operation, but the step forward is quantitatively moderate and I would not regard it as a breakthrough. (Also, one should bear in mind that reaching the 99% fidelity threshold is a necessary but practically insufficient condition for fault-tolerant quantum computing. The authors state this explicitly in the outlook section.)

In addition, I would like to point out that in the prospect of developing scalable quantum processors based on semiconductor qubits, establishing high-fidelity qubit operation at elevated temperatures is far from being the most important challenge.

Finally, the manuscript by Huang et al is densely written with many technical details. It would be suitable for a specialize journal but hardly accessible to the broad readership of Nature.

Author Rebuttals to Initial Comments:

Response to Reviewer 1

In this manuscript, the authors present an impressive set of spin qubit results at a temperature of 1K. The main new feature of this work is the improvement of two-qubit gate fidelity at 1K from previously reported values around 90% to a new record around 99%. This is not only important for eventual surface code viability, but it already enables a new algorithmic initialization procedure.

My technical comments are as follows.

We thank the reviewer for the supportive comments and helpful suggestions. Below we provide our responses to the questions.

1) On line 84 the authors refer to the valley splitting being smaller than the orbital excitation energy. It would be good to include those numbers for this device.

We thank the reviewer for the suggestion. These numbers would vary across different configurations, but as a guide for silicon MOS devices, orbital excitation is typically a few meV, and valley excitation is typically hundreds of μeV . We have added this and rephrased the sentence in question to be more explicit.

New text (line 75):

The precision needed to measure PSB is set by the excitation energy of the electrons in the dot, which depending on the Q1 or Q2 could consist of either the orbital excitation (typically a few meV) or a valley excitation (typically hundreds of μeV).

2) Related to the previous comment, why is a (1,3) charge state included in Figure 3? It seems that the authors have already stated that this should not be used due to potential valley excitations. And why is (3,3) not included?

We thank the reviewer for this pertinent question. It is known that the spin relaxation rate is similar for the 1-electron and 3-electron charge states at low temperatures (Fig. 3 in Yang *et al.*, *Nat Commun* 4, 2069 (2013). <https://doi.org/10.1038/ncomms3069>). Here, we study (1,3) and (5,3) for a broader insight into the physics at different temperatures, in addition to Ref. 16, which is based on (3,3).

We have added this information in the main text right after where Figure 3 is first mentioned.

New text (178):

We first study T_1 and T_2 (Figs. 3 a-b) in this device in the (1,3) and (5,3) charge states near the optimal B_0 . These regimes are expected to have similar relaxation mechanisms as (3,3) [51], which is studied in Ref. [16]. However, the absence of a micromagnet in the present study affects some of the physical mechanisms of relaxation and decoherence.

3) The abstract and introduction mention the one-qubit and two-qubit gate fidelities and SPAM fidelities being in the fault-tolerant range, but it seems to me this is slightly (unintentionally) misleading because the quoted fidelities are for different charge regimes. Figure 3 shows good one-qubit gate fidelities with either (1,3) or (5,3), while it appears from Supplementary Figure 6 that fidelities in (3,3) are much worse. Figure 4 gives good two-qubit gate fidelities for (3,3), but Extended Data Figure 8 shows that operating in (5,3) gives significantly worse quality factors. I could not find a specification of the charge regime the SPAM results were for. This should be clarified. In any case, it appears this device cannot achieve the fault-tolerant range in all aspects simultaneously at 1K.

We thank the reviewer for pointing this out and apologise for the ambiguity and missing information. We have added the charge configuration to Figure 2 and its description.

We have also rephrased the statements in the abstract and introduction to make it clearer that these aspects are not simultaneously above the fault-tolerant standard.

Updated Figure 2:

Fig. 2 | Initialisation and readout. a, Two-qubit algorithmic initialisation and the outcomes at $B_0 = 0.79$ T and 35 mT, both at $T = 1$ K in the (5, 3) charge configuration.

Modified section in abstract:

We also demonstrate single-qubit Clifford gate fidelities up to 99.85 per cent, and a two-qubit gate fidelity of 98.92 per cent.

Modified section in introduction (line 33):

In this work, we operate electron spin qubits in silicon with SPAM and universal logic fidelities approaching the requirements for surface code error correction [20-22, 24].

4) *In line 313, I don't think it is quite fair to say that RB is not reflecting the "true DCZ fidelity." The nonideal echo is part of the gate. It seems that what the authors have in mind as the "true" infidelity is the part that originates during the exchange pulses, excluding imperfections in the echo, and the FBT result can extract that. While the FBT analysis is of interest for diagnosing how to improve further, what is relevant for operation of a quantum processor is the infidelity of the full DCZ operation, which is here limited to around 98% by the single-qubit idle time during the echo. This is still an impressive improvement! I simply think it is confusing to report the FBT fidelity as the true fidelity, and it should be clarified that the RB fidelity is the more relevant one for fault tolerance.*

We thank the reviewer for the comment. We agree with the reviewer on their reasoning and have withdrawn the statement of "RB is not reflecting the true DCZ fidelity".

5) *In line 363 the authors mention the necessity of "advanced pulse shaping techniques to eliminate the non-Markovian noise sources." Some representative references here would be helpful.*

We thank the reviewer for suggesting this. We have cited some representative articles on several pulse engineering approaches.

Modified text (Supplementary Fig. 7 description):

It would require advanced pulse engineering techniques [49, 84–86] to eliminate the non-Markovian noise sources causing these inconsistencies.

6) *It would be useful for the authors to say what were the main factors in the improvement of 1K two-qubit fidelity compared to Ref. 16.*

We thank the reviewer for suggesting adding this explanation. We operate an improved device where the improvements target better noise performance. We have refined the gate deposition process and used better purified silicon substrate (50 ppm instead of 800 ppm in previous works). These improvements have led to less charge noise and magnetic noise respectively. Furthermore, there is no micromagnet in this device, unlike in Ref. 16. Without a micromagnet, charge noise only couples to the qubits through intrinsic spin-orbit coupling.

We have incorporated such information in the first paragraph in section A. Device and two-qubit operation.

New text (line 62):

We emphasise that these materials and experimental designs improve the noise performance of our qubits significantly in comparison to previous iterations [16]. For instance, the absence of magnetic materials in the design reduces the coupling of the spin to the electric noise generated by thermal fluctuations in dielectrics and metals.

In conclusion, this manuscript is a remarkable improvement in the state of the art, with exciting ramifications for the prospects of spin qubit scalability. I would recommend it for publication in Nature provided the comments above are satisfactorily addressed.

We thank the reviewer again for the positive appraisal, and we hope that the revision has addressed these issues.

Response to Reviewer 2

In "High-fidelity operation and algorithmic initialization of spin qubits above one kelvin", Huang et al. describe a novel protocol for two-qubit state preparation which they use to successfully prepare low entropy spin states that are far out of equilibrium with the local spin bath at 1K. Furthermore, they combine this initialization protocol with single- and two-qubit universal control and spin parity readout, all at 1K, to achieve near state-of-the-art fidelities comparable to recent spin qubit results operating at millikelvin temperatures.

In my opinion, the demonstration of a novel initialization protocol at temperatures approaching the single qubit energies, along with the successful execution of universal control in quantum dots, is a novel and outstanding result that I do think is worthy of publishing at Nature.

However, I do have some lingering issues that I would like the authors to address.

We thank the reviewer for the supportive comments and helpful suggestions. Below we provide our responses to the questions, in which we hope we have addressed all ambiguities.

1. In Fig. 1(c), a charge stability diagram is marked with a 'readout' operating point near the $(m, n+2) \rightarrow (m+1, n+1)$ charge transition, located about $VP1-VP2 \sim -0.1V$. But in Fig. 1(d), the author's show a 2D sweep of readout position vs blockade probability, swept over a narrow range about $VP1-VP2 \sim 0V$. Shouldn't the bias sweep of Fig.1(b) be near the bias conditions for the readout position in Fig.1(c)? And if not, I suggest that the author's clarify this point in the caption.

We thank the reviewer for pointing this out and apologise for the mistake in the axes. The bias sweep of Fig.1(c) is near the bias conditions for the readout position in Fig.1(b), and the original plots are in voltages relative to the DC bias points. We have added the DC offsets and updated the axes to absolute voltages.

Updated Fig. 1:

2. The author's state that they "prefer to load three electrons in at least one of the dots to avoid the small excitation energy" (line 82). However, there is no mention of expected valley energies in such configurations

We thank the reviewer for the pointing this out. These numbers would vary across different configurations, but as an estimate for SiMOS, orbital excitation is typically a few meV, and valley

excitation is typically hundreds of μeV . We have added this and rephrased the sentence to be more explicit. Please also see our response to Q1 of Reviewer 1.

New text (line 75):

The precision needed to measure PSB is set by the excitation energy of the electrons in the dot, which depending on the Q1 or Q2 could consist of either the orbital excitation (typically a few meV) or a valley excitation (typically hundreds of μeV).

and, given that the device is operating at elevated temperatures, there is no consideration for the probability of generating exciting states outside of the spin polarization basis that the author's use to describe their initialization protocol. I would imagine that their protocol should successfully filter out such excited valley states, as such states would not be blockaded by PSB readout and thus would be rejected in stage ii of the initialization protocol. Is this correct? In either case, I suggest that the authors should explicitly address the issue of active initialization in the (potential) presence of excited states. Such comments would not only strengthen the arguments of the paper but also relax any implicit assumptions for scalable high-fidelity initialization.

We thank the reviewer for the insightful consideration on this issue and the suggestion. The reviewer is correct that the excited states would not be blockaded by PSB readout and would be rejected in stage ii of the initialisation protocol. We have added this information to the description of the protocol.

Modified section in main text (line 99):

Fig. 2a shows the algorithmic initialisation protocol to initialise $|\downarrow\downarrow\rangle$ from a mixed state, and potentially in the presence of excited states.

Modified sections in Methods (lines 476, 494, 514):

3, Ramp to the PSB region for parity readout, and apply a filter that rejects odd-parity states. The parity readout preserves the even-parity states as long as it is performed faster than the spin relaxation time [35].

i, If the state is unblockaded and thus determined as an odd-parity ($|\downarrow\uparrow\rangle, |\uparrow\downarrow\rangle$) or excited state, the initialisation is restarted.

6, Ramp to the PSB region for parity readout, and apply a filter that rejects odd-parity states.

i, If the state is unblockaded and thus determined as $|\uparrow\downarrow\rangle$ or an excited state, the initialisation is restarted.

After Stage II, the output is a mixture of $|\downarrow\downarrow\rangle$ and $|\uparrow\uparrow\rangle$ with the odd-parity states or excited states filtered out through PSB, which can be identified from the associated ESR transitions.

3. On lines 221 – 223, “in this device, we expect the reduced charge and magnetic noise ...”, the authors state that the device used in this demonstration has improved noise performance. As far as I can tell, there is no discussion or explanation for such improvements. Please address the source(s) of such improvements.

We thank the reviewer for suggesting adding this explanation. We operate a better device focused on improvements that target better noise performance. We have refined the gate deposition process and used more purified silicon substrate (50 ppm instead of 800 ppm in previous works). These changes have led to less charge noise and magnetic noise, respectively. Furthermore, there is no micromagnet in this device, unlike the one in Ref. 16. Without a micromagnet, charge noise only couples to the qubits through intrinsic spin-orbit coupling.

We have incorporated such information in A. Device and two-qubit operation.

New text (line 62):

We emphasise that these materials and experimental designs improve the noise performance of our qubits significantly in comparison to previous iterations [16]. For instance, the absence of magnetic materials in the design reduces the coupling of the spin to the electric noise generated by thermal fluctuations in dielectrics and metals.

4. On lines 261-265, “... with the strong capability of noise decoupling [...] and fast spin rotations [...], the optimized single-qubit Clifford fidelity ...”, the authors imply that there is some amount of noise decoupling taking place in the gates. However, there is no explicit mention of single-qubit gate optimization in Methods F. Have the authors explicitly incorporate any explicit noise mitigation techniques here?

We thank the reviewer for this helpful question and agree that the phrasing that we have used in the sentence mentioned by the reviewer is not very clear. We have implemented several tweaks to the operation of the device that make it less sensitive to electric noise. We now list the noise mitigation measures and factors in the second paragraph of Methods F.

New text (line 634):

To optimise the single-qubit gate fidelity, we study different B_0 (Fig. 3 d) and tightly confine the qubits with low barrier gate voltages to reduce noise coupling. In single-qubit randomised benchmarking, the coherent driving decouples the qubit from noise to a certain extent [78], and the random rotations of the qubit also have the effect of refocusing [39, 79]. Here we optimise the microwave power and thus f_{Rabi} , such that the spins are driven quickly without excessive microwave-induced noise [45].

5. Please comment on the inconsistency between the quoted SPAM (readout + initialization) fidelity metrics of ~99%, compared to the full visibility of Rabi oscillation [Extended Data Fig. 6(f)], DCZ oscillations [Fig. 4(a)] and +Z/-Z RB decays (Supplementary Fig. 5), which all look to be <90%. I understand there are many reasons that could explain this difference (different device configurations,

tune-ups, or different sequences have predictably different visibility, etc) but an explanation from the author's would be very helpful.

We thank the reviewer for pointing this out and suggesting adding an explanation. In these measurements, the visibilities are indeed limited for various reasons. We have added such explanations in the descriptions of Extended Data Fig. 6(f), Fig. 4(a) and Supplementary Fig. 5.

New text (Extended Data Fig. 6(f) description):

Possible causes are off-resonance driving on the ancilla qubit, decoherence during off-resonance driving, or a greater impact from the microwave-induced noise [45]. All of these can contribute to the reduced oscillation amplitude.

New text (Fig. 4(a) description):

The visibility is limited by microwave-induced noise [45] and J gate pulsing, and the use of only partial algorithmic initialisation.

New text (Supplementary Fig. 5 description):

With this particular device configuration and the long physical gate sequences, the odd-parity readout fidelity is compromised.

6. Please comment on the use of fidelity "ratios" for reporting IRB fidelities. I am confused if this is referring to (1) the gate error deduced from the difference between interleaved and reference RB sequences, or (2) the actual ratio of sequence decay rates. I am familiar with reporting the interleaved gate fidelity directly, and less familiar with ratios. Perhaps this is my misunderstanding of notation, but if not, please cite any supporting material for the justification of IRB ratios.

We thank the reviewer for raising this question. We are reporting the interleaved gate fidelity, obtained from ratio of the interleaved sequence decay rate to the reference sequence decay rate. We have updated this term in the manuscript and provided a reference.

Modified section in main text (line 258):

We assess the DCZ gate metrics using two-qubit interleaved RB (IRB) [58] and fast Bayesian tomography (FBT) [61, 62] (see Methods), which report DCZ fidelities of 99.8 ± 0.2 %, 99.15 ± 0.13 % at $T = 0.1$ K, and 97.7 ± 1.5 %, 98.92 ± 0.67 % at $T = 1$ K.

Modified section in Methods (line 664):

From the two-qubit IRB decays, we first obtain an IRB fidelity [58] of 99.8 ± 0.2 % at $T = 0.1$ K, and 97.7 ± 1.5 % at $T = 1$ K, for the DCZ gate.

Response to Reviewer 3

The manuscript by Huang et al marks a step forward in the strive to increase the operation temperature of silicon spin qubits. Earlier works by the same research group and others had already provided some first proof-of-concept demonstrations showing a moderate loss of qubit performance when raising temperature from tens of mK to about 1K. Huang et al report further progress by introducing a new type of initialization procedure based on the Pauli spin blockade effect in combination with zCNOT conditional rotation. The achieved initialization fidelity exceeds 99% for the even-parity, two-qubit ground state.

The paper is complemented by a thorough characterization of one- and two-qubit gate fidelities, with results surpassing the current state-of-the-art. In particular, the fidelity of two-qubit, decoupled control-phase (DCZ) operations approaches 99%. The authors offer an extensive discussion of the possible causes of infidelity.

Overall, I find this work rich and scientifically sound. On a technical level, I have no serious criticism to raise. On the contrary, I am very impressed by the massive amount of meticulously performed, highly valuable work. The authors deploy a variety of sophisticated techniques, including the use of machine learning for error analysis.

We thank the review for the kind feedback and are glad that the value of this work is thoroughly recognised.

That said, this work is to my view unsuitable for publication in Nature. While it is certainly relevant to the spin qubit community, its novelty level lies well below the typical standards of a journal like Nature. The demonstrated initialization protocol is to a good extent original but I cannot view it as a clear breakthrough: 1) Pauli spin blockade is well known to work at temperatures well above the qubit characteristic energy (e.g., it was used in the earlier demonstrations of 1K spin-qubit operation by Yang et al. and Petit et al., ie refs 16 and 17, respectively); 2) Moreover, in refs 16 and 17, initialization fidelity was not measured and charge readout was performed by measuring current through a nearby SET, as opposed to the rf-SET readout technique used in this work. Therefore, it is difficult to judge the level of improvement with respect to those earlier works. The one- and two-qubit fidelities measured by Huang et al do exceed the current state-of-the-art for 1K operation, but the step forward is quantitatively moderate and I would not regard it as a breakthrough. (Also, one should bear in mind that reaching the 99% fidelity threshold is a necessary but practically insufficient condition for fault-tolerant quantum computing. The authors state this explicitly in the outlook section.)

We thank the reviewer for raising this concern. We view the incorporation of RFSET and the assessment of initialisation, control and readout fidelities all as improvements from Ref. 16 and Ref. 17, as they have never been tried until our work. The initialisation protocol leverages the high-fidelity readout provided by PSB and the RFSET, which represents a technical advance made possible by exploiting physical phenomena and modern engineering. The resulting fidelities are as good as the state-of-the-art at millikelvin temperatures, which neutralises the only drawback of operating at elevated temperatures. We acknowledge that further work is required to bring spin qubits towards

fault tolerance above 1 kelvin, and with this study, we hope to encourage and guide further developments through a popular journal like Nature.

In addition, I would like to point out that in the prospect of developing scalable quantum processors based on semiconductor qubits, establishing high-fidelity qubit operation at elevated temperatures is far from being the most important challenge.

We agree that there are many layers of challenges on the road to a scalable spin-based quantum computer – and the community may have only foreseen a few of them. For this reason, it may not be the most important challenge, but should be seen as a rather difficult one, as the elevated temperatures impose a hard barrier on all aspects of qubit performance. We hope that our understanding and work-around on these fundamental physics would convey a timely and inspiring story to the spin qubit community and beyond.

Finally, the manuscript by Huang et al is densely written with many technical details. It would be suitable for a specialize journal but hardly accessible to the broad readership of Nature.

We thank the reviewer for their view on the manuscript presentation. We have reorganised the contents to keep the higher-level and significant information, results, and discussions in the main text for the interest of Nature's general readership. We have moved the technical details to the relevant sections in Methods and Supplementary Information, which will still be useful to specialists from their respective fields.

Reviewer Reports on the First Revision:

Referees' comments:

Referee #1 (Remarks to the Author):

The authors have largely addressed my comments satisfactorily and I recommend publication in Nature.

However, I would urge the authors to be more precise regarding the orbital and valley excitation energies. I understand that these energies vary depending on the configuration, but stating that they are in the few meV/hundreds of ueV range is not informative enough. That statement alone could include cases as different as, e.g., 1meV/700ueV and 7meV/100ueV. It should not be difficult to include a supplemental table of measured valley splittings for each of the charge configurations discussed in this manuscript. If it varies substantially within a given charge configuration, it could even be broken down by the operating locations of Fig 1c. If this is not possible for some reason, at the very least a more detailed estimate would be helpful.

Referee #2 (Remarks to the Author):

I am pleased with the modifications that the author's have made and believe that they have addressed my concerns. I recommend that this article be published in Nature.

Referee #3 (Remarks to the Author):

The authors have made an effort to make their manuscript more readable. I find the first part quite accessible to a broad readership. Starting from the central part of page 3, however, the manuscript becomes progressively harder to follow.

I expect the non-specialist reader will lose interest in the second part, which is full of numbers on the performance and figures of merit of the two-qubit system. The novelty of the work lies to a good extent in these numbers. Therefore, I can very well see the difficulty of further improving readability without compromising the integrity of the message.

Numbers are clearly important and this work certainly marks a step forward in the prospect of developing high-fidelity spin qubits capable of operating around 1 K. Is this step relevant enough to justify publication in Nature? I do not think so.

The overall improvement in fidelities with respect to earlier demonstrations of high-temperature operation is clear, and it has certainly required a strong effort. But I regret to say that I would not

classify that as a breakthrough justifying publication in Nature:

1) reaching the 99% threshold is a necessary but not sufficient requirement for a realistic implementation of large-scale quantum computing based on surface-code quantum error correction. Besides that, the two-qubit fidelity barely approaches 99%.

2) the reported fidelity improvement refers to a specific spin qubit implementation (electrons in Si-MOS with ESR control via a local antenna). However, the scalability of this approach is questionable (indeed, in their outlook, the authors evoke the possibility of global field control as a more scalable route).

3) the reported initialization protocol adds another element of novelty. I feel the authors use a rather fancy name for what others would call real-time feedback control, which is not so new in the community (see e.g. Kobayashi et al., npj Quantum Inf 9, 52 (2023))

In conclusion, I confirm my initial judgement against publication in Nature, and I would rather recommend the publication of this work in a more specialized journal.

Some minor comments:

Line 118: dEz is not defined

Line 122: clarify the meaning of "pulsing exchange"

Line 191: "temperature dependence of all states" sounds weird.

Line 209: It should be " < 200 KHz"

Line 197-199: Could you speculate of the origin of the difference scaling power for the temperature dependence of T_1 and T_2 ?

Line 211-213: I would not be so sure about the statement that white noise originates from the qubit environment. Could it be a temperature dependent filtering of external noise instead?

Author Rebuttals to First Revision:

Response to reviewer comments

Reviewer 1

The authors have largely addressed my comments satisfactorily and I recommend publication in Nature.

However, I would urge the authors to be more precise regarding the orbital and valley excitation energies. I understand that these energies vary depending on the configuration, but stating that they are in the few meV/hundreds of ueV range is not informative enough. That statement alone could include cases as different as, e.g., 1meV/700ueV and 7meV/100ueV. It should not be difficult to include a supplemental table of measured valley splittings for each of the charge configurations discussed in this manuscript. If it varies substantially within a given charge configuration, it could even be broken down by the operating locations of Fig 1c. If this is not possible for some reason, at the very least a more detailed estimate would be helpful.

We thank the reviewer for this valuable suggestion to the improvement of our manuscript. We apologise for the lack of precision in the previous revision. We include the following table in Supplementary Information, which shows values of the valley and orbital excitation energies measured in previous experiments.

Device	Valley excitation	Orbital excitation
This work	-	2.5 meV
Ref. [1] (Device F)	0.2–0.3 meV	-
Ref. [2]	0.2–0.8 meV	~ 2 meV
Ref. [3]	~ 0.5 meV	2.9 meV
Ref. [4] (Device A)	-	2.6 meV

Supplementary Table I. Excitation energies.

In our work, we have access to the lowest excitation energy through our readout window, which is 2.5 meV. As we can see in the table, this energy is much larger than the valley excitation energies observed in our previous devices. Since we perform readout at the 3- to 4-electron transition, the addition energy of 2.5 meV required to overcome PSB corresponds to the orbital excitation energy, which is similar to those measured in our previous devices.

Confirming and extracting the valley excitation energy requires spin filling and magneto spectroscopy measurement [Yang *et al.*, *Nat. Commun.* **4**, 2069 (2013)], which would be time-consuming to perform in all configurations. As it is the orbital excitation energy – not the valley excitation energy – that limits the readout window, we did not consider the experiments necessary when our device was commissioned in the setup.

Modified main text (Line 105):

We perform readout at the 3- to 4-electron transition where the readout window spans 2.5 meV – much larger than the typical valley excitation energies and consistent with the orbital excitation energies previously observed in silicon shell filling (Supplementary Information).

Reviewer 2

I am pleased with the modifications that the authors have made and believe that they have addressed my concerns. I recommend that this article be published in Nature.

We thank the reviewer for the valuable contribution to the improvement of our manuscript.

Reviewer 3

Some minor comments:

Many thanks to the reviewer for these valuable reminders. We correct these issues as follows:

Line 118: dE_z is not defined

Now Line 147: we have added the definition “Zeeman energy difference” in front.

Line 122: clarify the meaning of “pulsing exchange”

Now Line 152: we have changed it to “pulsing on the J gate”.

Line 191: “temperature dependence of all states” sounds weird.

Now Line 221: we realise this error in phrasing, and we have changed it to “temperature dependence of T_1 displayed in Fig. 3 a”.

Line 209: It should be “< 200 kHz”

Now Line 242: we agree with the reviewer and have made the change.

Line 197–199: Could you speculate of the origin of the difference scaling power for the temperature dependence of T_1 and T_2 ?

We have added in Line 230: “We expect that the more purified silicon and the absence of a micromagnet in this study affects some of the physical mechanisms of relaxation and decoherence”.

This is also related to the part introducing the device in Line 91–97: “We emphasise that these materials and experimental designs improve the noise performance of our qubits significantly in comparison to previous iterations [16]. For instance, the absence of a micromagnet in the design reduces the coupling of the spin to the electric noise generated by thermal fluctuations in dielectrics and metals [31].”

Line 211–213: I would not be so sure about the statement that white noise originates from the qubit environment. Could it be a temperature dependent filtering of external noise instead?

We acknowledge this issue and have retrieved this speculation – a follow-up study on this would be helpful in confirming.